Contractile Function

# Obscurin deficiency leads to compensated dilated cardiomyopathy and increased arrhythmias

Josè Manuel Pioner[1]* , Enrico Pierantozzi[2]* , Raffaele Coppini[3] , Egidio Maria Rubino[2] , Valentina Biasci[4] , Giulia Vitale[5] , Annunziatina Laurino[2] , Lorenzo Santini[3] , Marina Scardigli[5] , Davide Randazzo[2] , Camilla Olianti[4] , Matteo Serano[2] , Daniela Rossi[2] , Chiara Tesi[5] , Elisabetta Cerbai[3] , Stephan Lange[6] , Carlo Reggiani[7,8] , Leonardo Sacconi[4] , Corrado Poggesi[5] , Cecilia Ferrantini[5] , and Vincenzo Sorrentino[2,9]

**Obscurin is a large muscle protein whose multiple functions include providing mechanical strength to the M-band and linking the sarcomere to the sarcoplasmic reticulum. Mutations in obscurin are linked to various forms of muscle diseases. This study compares cardiac function in a murine model of obscurin deletion (KO) with wild-type (WT) in vivo and ex vivo. Echocardiography showed that KO hearts had larger (+20%) end-diastolic and end-systolic volumes, reduced fractional shortening, and impaired ejection fraction, consistent with dilated cardiomyopathy. However, stroke volume and cardiac output were preserved due to increased end-diastolic volume. Morphological analyses revealed reduced sarcoplasmic reticulum volume, with preserved T-tubule network. While myofilament function was preserved in isolated myofibrils and skinned trabeculae, experiments in intact trabeculae revealed that *Obscn* KO hearts compared with WT displayed (1) reduced active tension at high frequencies and during resting-state contractions, (2) impaired positive inotropic and lusitropic response to β-adrenergic stimulation (isoproterenol 0.1 μM), and (3) faster mechanical restitution, suggesting reduced sarcoplasmic reticulum refractoriness. Intracellular $[Ca^{2+}]_i$ measurements showed reduced peak systolic and increased diastolic levels in KO versus WT cardiomyocytes. Western blot experiments revealed lower SERCA and phospholamban (PLB) expression and reduced PLB phosphorylation in KO mice. While action potential parameters and conduction velocity were unchanged, β-adrenergic stimulation induced more frequent spontaneous $Ca^{2+}$ waves and increased arrhythmia susceptibility in KO compared with WT. Taken together, these findings suggest that obscurin deletion, in adult mice, is linked to compensated dilated cardiomyopathy, altered E-C coupling, impaired response to inotropic agents, and increased propensity to arrhythmias.**

## Introduction

Titin, nebulin, and obscurin are three giant sarcomeric proteins that play critical structural and functional roles in striated muscle cells, including providing muscle elasticity, responding to stretch, and organizing the sarcomere. A detailed investigation of the functions of these proteins has led to the understanding that the sarcomere is not merely a force-generating structure; rather, it is now envisioned as a molecular hub where

multiple inputs such as mechanical stress, passive stretch, hormone signals, and metabolite availability are sensed and integrated (Gautel and Djinovic-Carugo, 2016; Wang et al., 2021; Rossi et al., 2022).

Obscurin is the most recently identified component of this group of giant proteins (Bang et al., 2001; Young et al., 2001). Owing to its modular organization that contains multiple

[1]Department of Biology, University of Florence, Florence, Italy; [2]Department of Molecular and Developmental Medicine, University of Siena, Siena, Italy; [3]Department NeuroFarBa, University of Florence, Florence, Italy; [4]Institute of Clinical Physiology, National Research Council (IFC-CNR), Florence, Italy; [5]Department of Experimental and Clinical Medicine, University of Florence, Florence, Italy; [6]Institute of Biomedicine, Aarhus University, Aarhus, Denmark; [7]Department of Biomedical Sciences, University of Padua, Padua, Italy; [8]Science and Research Center Koper, Institute for Kinesiology Research, Koper, Slovenia; [9]Program of Molecular Diagnosis of Rare Genetic Diseases, Azienda Ospedaliero Universitaria Senese, Siena, Italy.

*J.M. Pioner and E. Pierantozzi contributed equally to this paper.   Correspondence to Vincenzo Sorrentino: vincenzo.sorrentino@unisi.it;   Cecilia Ferrantini: cecilia.ferrantini@unifi.it

D. Randazzo's current affiliation is Light Imaging Section, National Institute of Arthritis and Musculoskeletal and Skin Diseases, National Institutes of Health, Bethesda, MD, USA.

This work is part of a special issue on Myofilament Structure and Function.

protein binding sites and signaling domains, obscurin contributes to the scaffolding mechanism for the overall architectural arrangement of the sarcomere and provides a functional interaction in maintaining the sarcoplasmic reticulum (SR) around the contractile filaments, and finally, it is actually envisioned as one of the signaling focal points that are involved in striated muscle functions (Bagnato et al., 2003; Kontrogianni-Konstantopoulos et al., 2009; Gautel and Djinović-Carugo, 2016; Rossi et al., 2022).

The *OBSCN* gene encodes two giant isoforms in adult striated fibers: obscurin A and B. These isoforms are primarily located at the M-band, with a minor presence at the Z-disk (Bowman et al., 2007; Borisov et al., 2008). Both share a modular N-terminal region with Ig-like and Fn3-like domains, enabling them to bind to titin and myomesin, and slow myosin binding protein C at the M-band and titin at the Z-disk. These complexes contribute to M-band lattice organization and sarcomere stability. Only giant obscurin isoforms bind titin at both the M-band and the Z-disk, helping to scaffold sarcomere boundaries (Fukuzawa et al., 2005, 2008; Ackermann et al., 2009; Meyer and Wright, 2013; Pernigo et al., 2015). Additionally, both isoforms contain signaling domains, including a Rho-GEF domain, a calmodulin IQ binding motif, a PH domain, and an SH3 domain, which may support myofibrillogenesis, fiber growth, and differentiation (Fukuzawa et al., 2005; Bowman et al., 2008; Coisy-Quivy et al., 2009; Shriver et al., 2016; Randazzo et al., 2017b).

Obscurin A and B differ in their C-terminal domains: obscurin A has an ankyrin binding domain that interacts with sAnk1.5 to tether the SR to the contractile apparatus and with ankyrin-B, linking to the dystrophin complex (Bagnato et al., 2003; Kontrogianni-Konstantopoulos et al., 2003; Armani et al., 2006; Lange et al., 2009; Busby et al., 2010; Giacomello et al., 2015; Rossi et al., 2022; Cunha and Mohler, 2008; Randazzo et al., 2013). Ankyrin-B also plays a key role in cardiac physiology, and mutations in ANKB are linked to cardiac arrhythmias and to other structural forms of heart disease (Mohler et al., 2003; El Refaey and Mohler, 2017; Sucharski et al., 2020). The C-terminal domain of obscurin B contains two serine/threonine kinase–like domains, also found in shorter cardiac obscurin isoforms (Russell et al., 2002; Fukuzawa et al., 2005; Grogan et al., 2020). These kinase domains share homology with the SPEG kinase and are part of the myosin light-chain kinase family, though their substrates remain unidentified (Hu and Kontrogianni-Konstantopoulos, 2013).

Shorter *OBSCN* transcripts, likely from exon shuffling and alternative polyadenylation, have been found in various tissues and are being studied for their role in cancer progression (Guardia et al., 2021).

An *OBSCN*-related gene, obscurin-like1 (*OBSL1*), encodes smaller isoforms with a modular organization similar to obscurin's N-terminal Ig- and Fn3-like domains (Geisler et al., 2007). Although sharing functional roles with obscurin, OBSL1 can only interact with myomesin and titin at the M-band and lacks the Z-disk binding sites and signaling domains found in obscurin (Blondelle et al., 2019).

Several *OBSCN* variants with pathological relevance to skeletal muscle tissues have been identified, including the R4444W missense mutation in the titin-interacting Ig59 domain in a family presenting with distal muscular dystrophy (Rossi et al., 2017) and loss-of-function *OBSCN* variants linked to rhabdomyolysis and altered Ca²⁺ handling (Cabrera-Serrano et al., 2022; Zemorshidi et al., 2024). However, since these patients did not exhibit overt cardiomyopathy, heart involvement was not investigated. Several studies have identified *OBSCN* variants in patients with cardiomyopathies, including hypertrophic cardiomyopathy, dilated cardiomyopathy (DCM), and left ventricular non-compaction, though their roles are still under investigation. In vitro studies initially suggested a potential causative role of the R4344Q (Arimura et al., 2007) and R4444W (Rossi et al., 2017) *OBSCN* variants, due to reduced interaction with titin. However, other studies did not support these findings (Fukuzawa et al., 2021). This discrepancy may be due to undetected effects of these mutations in in vitro experiments or may reflect the involvement of additional proteins. Furthermore, as these mutations were found alongside other variants (A4484T in *OBSCN* and G1722ValfsTer61 in *FLNC*), the observed phenotypes may result from the combined effects of multiple pathogenic variants.

The causative role of obscurin variants in human striated muscle diseases remains debated (Arimura et al., 2007; Marston et al., 2015; Xu et al., 2015; Rowland et al., 2016; Marston, 2017; Rossi et al., 2017; Grogan and Kontrogianni-Konstantopoulos, 2019; Fukuzawa et al., 2021; Noureddine and Gehmlich, 2023). However, animal models provide evidence supporting obscurin involvement in disease. Three genetically modified obscurin mouse models have been developed. The Obscn knockout (KO) mouse (Lange et al., 2009) showed shortened longitudinal SR and reduced SR Ca²⁺ release in skeletal muscle fibers, also observed in the sAnk1.5 KO model (Giacomello et al., 2015). In *Obscn* KO mice, obscurin absence disrupted ankyrin-2–dependent dystrophin localization, causing sarcolemma fragility and microtubule disarrangement after repeated contractions (Randazzo et al., 2013). Skeletal muscle fibers also showed M-band loss and Z-disk streaming after exercise (Randazzo et al., 2017a). However, no cardiac phenotype was reported in *Obscn* KO mice, as most studies focused on skeletal muscle.

The pathophysiological role of obscurin in the heart has been studied in a knock-in murine model carrying the R4344Q OBSCN variant linked to hypertrophic cardiomyopathy (Arimura et al., 2007; Guardia et al., 2021). This mutation affects the Ig58 domain, which, along with the Ig59 domain, interacts with titin at the Z-disk (Rowland et al., 2016; Grogan and Kontrogianni-Konstantopoulos, 2019; Noureddine and Gehmlich, 2023). Homozygous R4344Q mice developed arrhythmia, heart disease, and disrupted Ca²⁺ handling (Hu et al., 2017; Hu and Kontrogianni-Konstantopoulos, 2020). Similar cardiac dysfunctions were observed in a separate mouse model with deletions in the Ig58 and Ig59 domains (Grogan et al., 2020a). It was suggested that Ca²⁺ deregulation in both models may be linked to altered phospholamban (PLB) binding by obscurin. However, whether these effects are due to defective interactions or conformational changes in the mutant proteins remains unclear (Fukuzawa et al., 2021). A recent study also

reported diastolic dysfunction and deregulated metabolism and mitophagy in a double knockout of Obscn and Obsl1 (Fujita et al., 2022, *Preprint*). However, no previous study has fully determined how the absence of obscurin alone affects cardiac function. This highlights the need for a comprehensive evaluation of obscurin role in the heart. Here, we report data collected from in vivo and ex vivo experiments performed with *Obscn* KO mice.

## Materials and methods

### Animals
All the procedures that include the use of animals were performed in compliance with the directive 2010/63/EU of the European Parliament and the Council of 22 September 2010 about welfare of animals used for scientific purposes, and the study is reported in accordance with ARRIVE guidelines (https://arriveguidelines.org). 8-mo-old control (wild type [WT]) and *Obscn* KO Black Swiss male mice were used in the experimental protocols, in agreement with the current Italian and European regulations. All procedures were performed following the approval of the Animal Care Committee of the University of Siena and of the University of Florence, and following the approval and authorization of the Italian Ministry of Health. *Obscn* KO mice were previously generated by Professor Chen's group (Lange et al., 2009). Mouse colony was maintained by intercrossing heterozygous mice, which were housed at room temperature of 21–25°C, relative humidity of 50–60%, and dark–light cycle of 12 h, with free access to food and water. Mouse genotype was determined by PCR, as previously described (Randazzo et al., 2013). The mouse colonies were housed in the animal facility of the University of Siena.

### In vivo studies
Echocardiography was performed on isoflurane-anesthetized mice as previously described (Pistner et al., 2010) to characterize left ventricular (LV) morphology, and systolic and diastolic function, using B-mode imaging and Doppler measurements of transmitral blood flow. In detail, we performed echocardiographic measurements in collaboration with Toscana Life Sciences (Siena, Italy). We used the dedicated instrumentation VisualSonics Vevo 2100. Mice were first anesthetized in an induction chamber with isoflurane (induction, 4–5%; maintenance, 1–2.5% in oxygen from a precision vaporizer) and then positioned prone on a heated bed. In each mouse, we performed parasternal long axis and measured the thickness of the interventricular septum, left ventricle posterior wall, left ventricular interior diameter, short axis, Simpson's LV volume reconstruction, PV flow, LV diastolic function, and isovolumic relaxation time (IVRT), isovolumic contraction time, no flow time, and aortic ejection time.

### Isolated and perfused mouse hearts
WT and *Obscn* KO mice were heparinized (0.1 ml at 5,000 U/ml) and anesthetized by inhaled isoflurane (5%). The excised heart was immediately bathed in Krebs–Henseleit (KH) solution and cannulated through the aorta. The ratio between cardiac mass and body mass was calculated. The KH buffer contained (in mM)

120 NaCl, 5 KCl, 2 $Mg_2SO_4$-$7H_2O$, 20 $NaHCO_3$, 1.2 $NaH_2PO_4$-$H_2O$, and 10 glucose, pH 7.4, when equilibrated with carbogen (95% $O_2$–5% $CO_2$). The contraction was inhibited for the entire experiment with blebbistatin (5 μM) in the solution. The cannulated heart was retrogradely perfused (Langendorff's perfusion) with the KH solution and then used for three alternative purposes.

In the first subset of experiments, thin unbranched trabeculae were dissected for intact muscle mechanical studies; the remaining myocardium was treated with Triton to obtain skinned muscle strips/myofibrils to assess sarcomere mechanics and energetics or frozen at –80°C (Ferrantini et al., 2017).

In a second subset of experiments, the perfused hearts were dedicated to single cardiomyocyte isolation through enzymatic digestion for intracellular $Ca^{2+}$ fluxes and T-tubule (TT) density studies. In a third subset, the perfused hearts were employed for optical mapping studies in the Langendorff-perfused whole hearts.

### Myocardial mechanics from single myofibrils to skinned and intact multicellular preparations
Single myofibrils were isolated from ventricular samples and used for mechanical measurements at 15°C using a fast solution-switch technique to assess maximal $Ca^{2+}$-activated tension, resting tension (RT), and the kinetics of active tension generation and relaxation (Belus et al., 2008, 2010). Skinned trabeculae were employed to obtain pCa–tension curves as previously described (Chandra et al., 2001). Sarcomere energetics was assessed in skinned trabeculae by simultaneous measurement of isometric force and ATPase activity with an enzyme-coupled assay at 20°C (Witjas-Paalberends et al., 2014a, 2014b).

LV intact trabeculae were dissected from explanted hearts (Ferrantini et al., 2014, 2016) and used to record isometric force during electrical stimulation with different pacing protocols, at baseline and following β-adrenergic activation at 37°C (Crocini et al., 2016; Ferrantini et al., 2017).

### Assessment of excitation–contraction coupling morphofunctional remodeling
Single cardiomyocytes were isolated from excised hearts via enzymatic dissociation and used for intracellular $Ca^{2+}$ measurements using the Fura-2 $Ca^{2+}$-sensitive fluorescent dyes (Crocini et al., 2016; Ferrantini et al., 2016) to evaluate the amplitude and kinetics of $Ca^{2+}$ transients, diastolic $[Ca^{2+}]$, and the rate of spontaneous $Ca^{2+}$ release during stimulation with field electrodes (Coppini et al., 2017). To assess TT density, myocytes were stained with membrane-selective dyes and observed with a confocal microscope. Briefly, cardiomyocytes were stained by adding to the cell suspension 2 μg/ml of the voltage-sensitive dye (VSD) di-3-ANEPPDHQ (dissolved in ethanol). After washing, cells were resuspended in fresh $Ca^{2+}$-free solution containing 10 μM blebbistatin and 4 μM cytochalasin D. Loaded preparations were used for experiments within 30 min. The staining and imaging sessions were performed at room temperature (20°C), as previously reported (Crocini et al., 2016; Coppini et al., 2017).

## Voltage mapping in the Langendorff-perfused heart

WT and Obscn KO mice were heparinized via intraperitoneal injection (0.1 ml, 5,000 U/ml) and anesthetized with 5% inhaled isoflurane. The heart was excised, while the mouse was in deep terminal anesthesia, immediately cannulated at the aorta, and perfused with the KH buffer solution ([in mM]: 120 NaCl, 5 KCl, 2 MgSO$_4$·7H$_2$O, 20 NaHCO$_3$, 1.2 NaH$_2$PO$_4$·H$_2$O, 10 glucose). The solution was adjusted to pH 7.3 after equilibration with carbogen (5% CO$_2$, 95% O$_2$). Cardiac contractions were inhibited by adding 10 µM (±) blebbistatin to the solution.

The cannulated heart was placed in a custom-built optical mapping chamber (the horizontal Langendorff mode) and perfused at a constant flow rate of 2.5 ml/min at 37°C. Next, 1 ml of perfusion solution containing the VSD (di-4-ANBDQPQ; 8 µg/ml) was injected into the aorta. Whole mouse hearts were imaged at a frame rate of 1 kHz (with 1-ms actual exposure time) using a custom-made optical mapping system as previously described (Scardigli et al., 2018; Biasci et al., 2022). Optical recordings of the LV free wall were collected while pacing the heart with 15 pulses at 5 Hz. Susceptibility to cardiac arrhythmias was assessed by electrically pacing the heart at the right ventricular outflow tract using 40-Hz bursts (10 trials of 400 pulses each) in the presence of the β-adrenergic agonist isoproterenol (Iso) at $10^{-7}$ M, as previously reported (Scardigli et al., 2020).

## Histological analysis

8-mo-old mice were heparinized (0.1 ml at 5,000 U/ml) 15 min prior to euthanasia. The excised heart was immediately bathed and cannulated through the aorta and perfused with 30 ml of cooled phosphate-buffered saline (PBS) at a constant pressure of about 10 ml/min to remove blood. Finally, the heart was perfused with 20 ml of a cold solution containing 4% paraformaldehyde in PBS and incubated overnight in the same solution at 4°C. The next day, the heart was washed extensively in PBS at 4°C, embedded in OCT, and cooled in isopentane/nitrogen. OCT-embedded hearts were sectioned at a thickness of 8 µm using a cryostat (Leica) maintained at –20°C. Sections were stained with hematoxylin/eosin as previously described (Vattemi et al., 2022), and images were acquired at low magnification with a 2× objective of an FV3000 Olympus microscope (Carl Zeiss). Immunofluorescence staining was performed as previously described (Barone et al., 2015) by using a polyclonal antibody against collagen III (ab7778; Abcam), which was revealed by Alexa Fluor 488–conjugated anti-rabbit secondary antibody (A32731; Invitrogen). Images were acquired by an LSM 510 META confocal microscope equipped with a Plan-Neofluar 63× objective (Carl Zeiss).

EM samples were prepared and analyzed as previously described (Lange et al., 2009). Briefly, hearts were perfused with heparin followed by 30 mM KCl solution and subsequently by fixative (2% paraformaldehyde, 2% glutaraldehyde in 0.15 M sodium cacodylate buffer, pH 7.4) through the LV. The ventricular free wall was cut into smaller pieces and kept in fixative for 4 h. Postfixation was done overnight with a 0.15 M sodium cacodylate buffer containing 1% OsO$_4$, 0.8% potassium ferrocyanide. Subsequently, tissue pieces were stained for 2 h in 2% uranyl acetate, dehydrated, and embedded into Durcupan resin

(EMD). Ultra-thin sections (60–70 nm) were stained with uranyl acetate and lead citrate solution and imaged with a JEOL 1200EX transmission electron microscope at an accelerating voltage of 80 kV.

## Western blot analysis

The LV from WT and Obscn KO mice were homogenized in RIPA buffer (Cell Signaling Technology) containing 1 mM phenyl-methanesulfonyl fluoride (Merck KGaA). Total protein levels were quantified using the Pierce Protein Assay/bicinchoninic acid assay (Thermo Fisher Scientific). Proteins were separated on 4–20% SDS-PAGE (Thermo Fisher Scientific) and transferred onto a nitrocellulose membrane as in Rossi et al. (2020). Membranes were blocked with 5% nonfat dry milk in Tris-buffered saline with 0.1% Tween-20 (TBS-T) (Cell Signaling Technology) or 5% BSA in TBS-T for 1 h at room temperature, and then incubated overnight at 4°C with the following primary antibodies: anti-SERCA TRY2 used at 1:1,500 (Mountian et al., 2001), anti-phospho-PLB (Ser16) used at 1:1,000 (cat. 07-052; Merck KGaA), anti-PLB used at 1:1,000 (cat. PA5-26004; Thermo Fisher Scientific), anti-GAPDH used at 1:2,000 (MA5-15738; Thermo Fisher Scientific) diluted in 5% nonfat dry milk or 5% BSA in TBS-T. Blots were washed with TBS-T and then incubated for 1 h at room temperature with HRP-conjugated secondary antibodies (anti-rabbit IgG, HRP-linked antibody, cat. 7074, anti-mouse IgG, HRP-linked antibody, cat. 7076; Cell Signaling Technology) diluted 1:2,000 or 1:3,000 in 5% BSA or 5% milk in TBS-T solution. Washed membranes were incubated with enhanced chemiluminescence solutions (ECL; Bio-Rad) for 5 min, and the luminescent signal was acquired by a ChemiDoc luminescence counter (Bio-Rad). For total PLB analysis, phospho-PLB(Ser16) membrane was stripped at 56°C for 20 min in 10 ml of stripping solution (SDS 2%, Tris–HCl 0.5 M, 2-mercaptoethanol 0.08 ml [Merck KGaA]). Quantification of the intensities of immunoreactive bands was performed by ImageLab software (Bio-Rad), using the optical density of GAPDH as a normalizer.

## Statistical analysis

Data from myofibrils, cells, and muscles are expressed as the mean ± SEM (the number of samples and animals are indicated in the respective figure legends). Statistical analysis was performed using SPSS 23.0 (IBM) and STATA 12.0 (StataCorp). The statistical tests used to calculate P values for each dataset are indicated in the respective figure legends. For variables where a single measurement for each mouse is included (e.g., echocardiography, western blot), Obscn KO and WT were compared using one-way ANOVA with Tukey's correction (for normally distributed homoskedastic datasets). Overall, $P < 0.05$ was considered statistically significant. The range of calculated P values for each comparison $(0.05 > P > 0.01, 0.01 > P > 0.001)$ is indicated in the respective figure panels using symbols.

# Results

## Compensated DCM phenotype in Obscn KO hearts

Echocardiographic measurements were performed on anesthetized WT and Obscn KO 8-mo-old male mice, using a

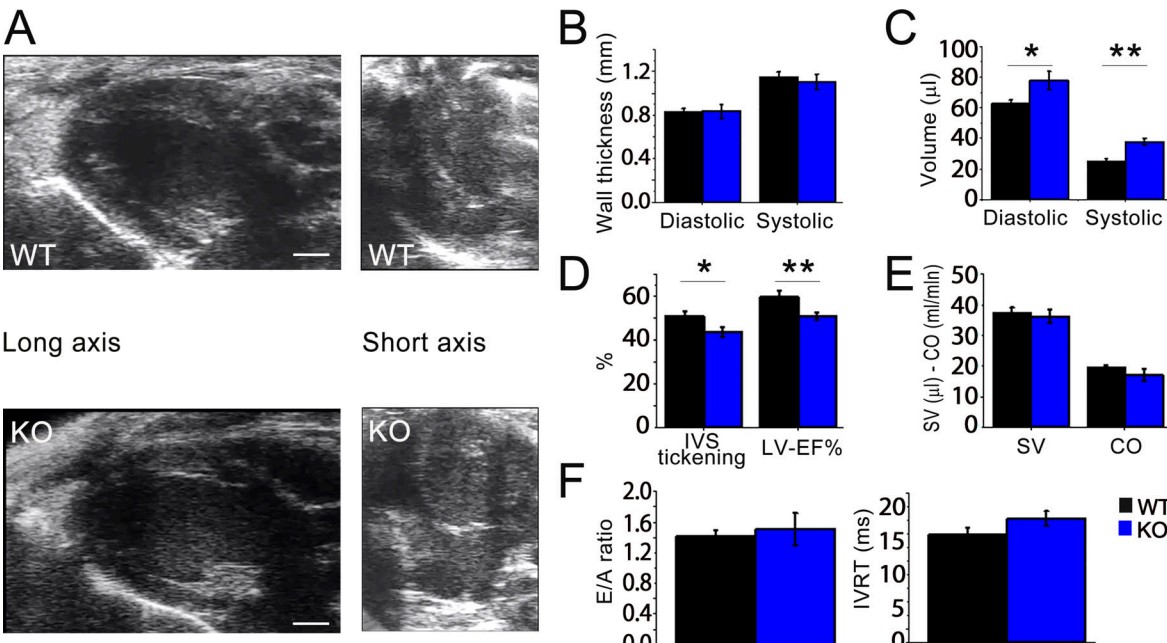

Figure 1. **Echocardiographic measurements. (A)** Representative parasternal long-axis and short-axis views of the LV at end-diastole from 8-mo-old WT and *Obscn* KO male mice. Horizontal scale bars equal 1 mm. **(B)** Thickness of the IVS measured at end-diastole (left) and at end-systole (right) in WT and *Obscn* KO mice. **(C)** LV volumes calculated using the Simpson technique at end-diastole (left) and end-systole (right) in mice from the two study groups. **(D)** Systolic thickening of the IVS (left) and LV ejection fraction (LV-EF%, right), expressed as the percentage of the diastolic values, measured in mice from the two cohorts. **(E)** SV calculated from the Simpson volumes and CO. **(F)** Ratio between E and A waves (E/A ratio, left) and isoVolumic relaxation time (IVRT, right) from transmitral blood flow velocity curves recorded using pulsed-wave Doppler echocardiography in apical views. Statistical tests: One-way ANOVA with Tukey's correction. Data are means ± SEM from 7 *Obscn* KO and 10 WT mice, respectively. *0.05 > P > 0.01; **0.01 > P > 0.001. IVS, interventricular septum; SV, stroke volume; CO, cardiac output.

standardized protocol (Merx et al., 2014). The mean spontaneous heart rate in sinus rhythm during echocardiography was 480–540 bpm (8–9 Hz), with no differences between the two groups. Representative images of parasternal long- and short-axis views of the LV are shown in Fig. 1 A. LV wall thicknesses were preserved in *Obscn* KO hearts (Fig. 1 B).

Compared with controls, *Obscn* KO hearts operated at significantly higher ventricular volumes, i.e., approximately +20% end-diastolic volume (EDV) and end-systolic volume (Fig. 1 C).

In *Obscn* KO mice, the ejection fraction (LV-EF%) was slightly but significantly reduced compared with WT controls (Fig. 1 D), suggesting reduced contractility. An additional sign in this direction was the reduced wall thickening during contraction, estimated at the interventricular septum (Fig. 1 D).

However, as *Obscn* KO hearts started LV ejection from elevated EDVs, we found that LV stroke volume was maintained, despite the reduced EF% (Fig. 1 E). Consistent with preserved stroke volume, cardiac output was maintained (with no differences in heart rate, as in the echocardiographic measurements) (Fig. 1 E). No evidence of diastolic dysfunction was observed in *Obscn* KO hearts. Specifically, Doppler studies of transmitral blood flow velocity (Fig. 1 F, left) were performed in four-chamber views and transmitral blood flow velocities during early diastole and atrial contraction (hence the E/A ratio) were similar in *Obscn* KO and WT mice. Furthermore, the LV IVRT was normal in *Obscn* KO (Fig. 1 F, right). Taken together, these

echocardiographic measurements indicate that the two hallmarks of the human DCM phenotype, i.e., LV dilatation and LV hypocontractility, are reproduced by this *Obscn* KO mouse model, but the reduced contractility is mild and counteracted by the increased EDV, resulting in a compensated DCM phenotype.

The heart weight of *Obscn* KO mice was unchanged (Fig. 2, A and B), and histological analysis did not reveal changes in walls and septum thicknesses, in line with the echocardiographic measurements. Furthermore, the analysis of type III collagen expression through immunofluorescence indicated that no interstitial fibrosis was present in the hearts of *Obscn* KO mice (Fig. 2 C). To detect potential changes in the organization of TT, isolated cardiomyocytes were labeled with the membrane-selective dye Di-3-ANEPPDHQ and imaged with a confocal microscope. TT profile was then quantified with dedicated software based on the fast Fourier transform (Ferrantini et al., 2014). As shown in Fig. 2, D and E, TT density and distribution were preserved in *Obscn* KO mice. However, analysis of electron micrographs taken from WT and KO hearts revealed ultra-structural changes in both longitudinal SR and TT area that resembled those observed in *Obscn* KO skeletal muscles (Lange et al., 2009). Specifically, the longitudinal SR in *Obscn* KO hearts showed a significant reduction in extension (Fig. 2, F and G). Additionally, *Obscn* KO hearts exhibited more disorganized junctional SR, accompanied by noticeable TT swelling, indicated by a significantly increased TT area compared with WT hearts (Fig. 2 H).

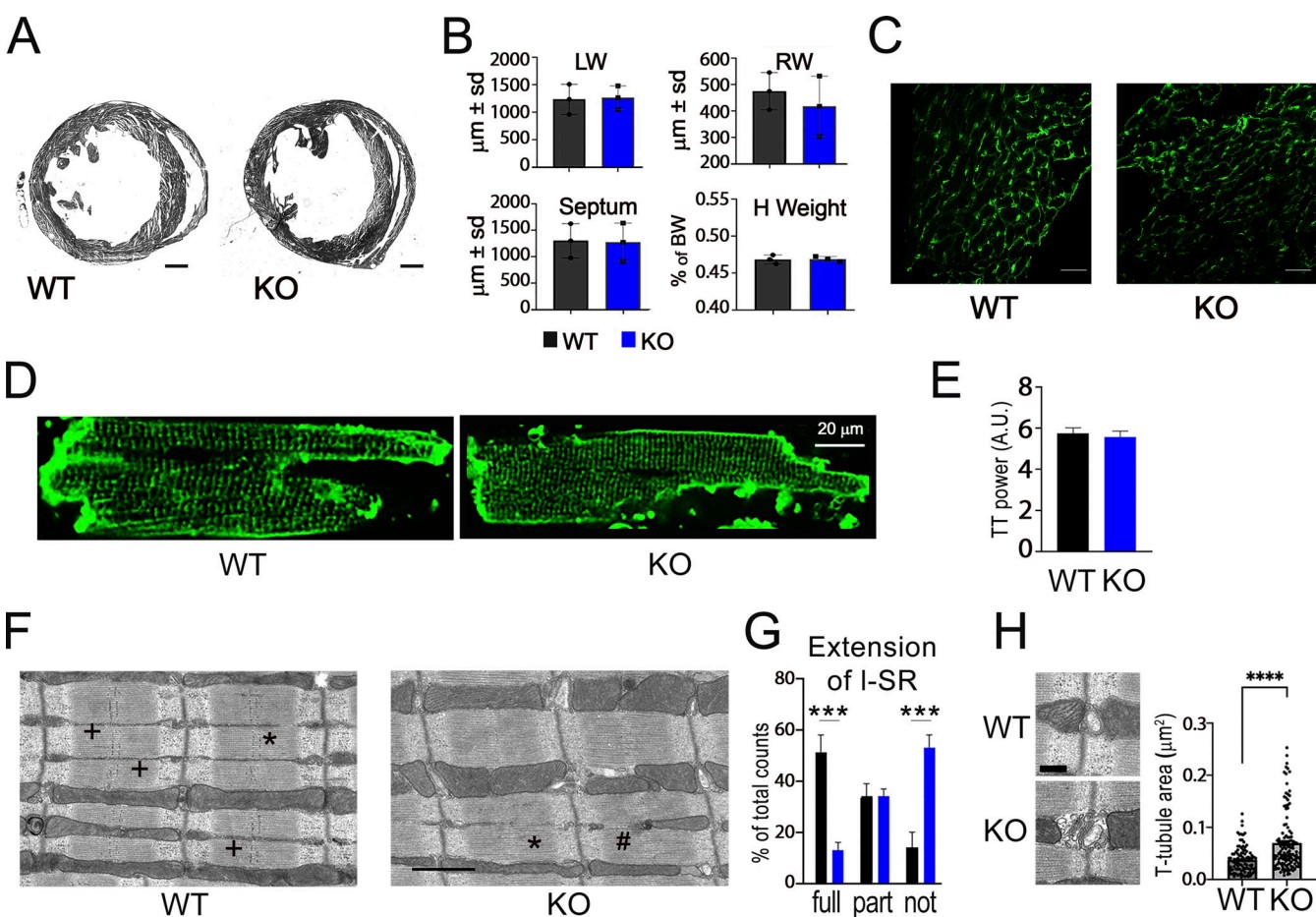

Figure 2. **Histological analyses. (A)** Representative hematoxylin and eosin staining of cross sections from 8-mo-old WT and *Obscn* KO hearts (*n* = 3). Bar = 1 mm. **(B)** Thickness (μm ± SD) of left and right walls, and of the interventricular septum of WT and *Obscn* KO hearts. Heart weight, expressed as the percentage of body weight, is shown in the bottom right panel. **(C)** Representative image of immunofluorescence staining for the detection of collagen III (scale bar equals 1 mm). **(D)** Representative confocal images from isolated LV cardiomyocytes stained with Di-3-ANEPPDHQ. The horizontal bar equals 20 μm. **(E)** TT density, as calculated using the Ttorg ImageJ plug-in, in cardiomyocytes from the two groups. **(F)** Representative electron micrographs of WT and *Obscn* KO hearts. +, *, and # indicate fully, partial, and not extended l-SR, respectively. Scale bar = 1 μm. **(G)** Analysis of l-SR extension in WT and *Obscn* KO myofibrils. Quantification of fully (full), partial (part), and not extended l-SR in WT and *Obscn* KO hearts is reported as the percentage for each type of l-SR. ***P < 0.001 versus WT, as determined by Student's *t* test. **(H)** Representative electron micrographs of WT and *Obscn* KO TT and analysis of TT area in WT and *Obscn* KO hearts. ****P < 0.0001 versus WT, as determined by a two-tailed Mann–Whitney test. Scale bar = 250 μm. l-SR, longitudinal SR.

### Sarcomere mechanics and energetics are preserved in *Obscn* KO myofibrils and skinned preparations

Ventricular myofibrils from *Obscn* KO and WT hearts, mounted for force recording at 15°C with optimum myofilament overlap, were maximally $Ca^{2+}$-activated (pCa 4.5) and fully relaxed (pCa 9.0) by rapid solution switching, as described previously (Belus et al., 2008; Ferrantini et al., 2017). No changes in maximal $Ca^{2+}$-activated tension (Po), RT, rates of tension generation and tension redevelopment ($k_{ACT}$ and $k_{TR}$), and in addition kinetic parameters of tension and relaxation were observed between *Obscn* KO and WT myofibrils (Table 1, upper part).

The effect of obscurin absence on sarcomere energetics at rest and during contraction was also studied by simultaneous measurement of isometric tension and ATPase activity in skinned ventricular trabeculae at 20°C (Table 1, lower part). Maximal $Ca^{2+}$-activated tension and RT were not different compared with controls. Importantly, both resting and maximal $Ca^{2+}$-activated ATPases were unchanged in *Obscn* KO skinned

trabeculae compared with WT preparations. Mean pCa-active tension curves, and thus myofilament $Ca^{2+}$ sensitivity (pCa50), were also not different.

### Impaired contractility and reduced lusitropic response to β-adrenergic stimulation in intact *Obscn* KO trabeculae

Despite the absence of sarcomeric changes, intact *Obscn* KO trabeculae showed reduced active tension at high frequencies and during rested-state contractions, as well as impaired positive inotropic and reduced lusitropic responses to β-adrenergic stimulation. Specifically, isometric force was measured from intact left and right ventricular trabeculae or thin papillary muscles during field stimulation at 0.1–8 Hz, 30°C, and 2 mM extracellular [$Ca^{2+}$], in the absence and presence of Iso 0.1 μM. Twitch contraction measurements at baseline and under Iso treatment were used to mimic β-adrenergic stimulation. The amplitude of the baseline twitch contraction was similar in *Obscn* KO and WT trabeculae at 1–2 Hz (Fig. 3, A and B), but was

**Table 1. Mechanical and energetic parameters of myofibrils and skinned trabeculae from WT and *Obscn* KO mice**

| Myofibril | RT | Po | $k_{ACT}$ | $k_{TR}$ | $D_{slow}$ | Slow $k_{REL}$ | Fast $k_{REL}$ |
|---|---|---|---|---|---|---|---|
| | mN mm$^{-2}$ | mN mm$^{-2}$ | s$^{-1}$ | s$^{-1}$ | ms | s$^{-1}$ | s$^{-1}$ |
| WT (N = 5) | 11 ± 1.3 (n = 19) | 108 ± 12 (n = 27) | 9.1 ± 0.69 (n = 32) | 8.6 ± 0.92 (n = 24) | 78.5 ± 6.0 (n = 19) | 1.62 ± 0.28 (n = 19) | 25 ± 3 (n = 19) |
| *Obscn* KO (N = 3) | 14.6 ± 2.3 (n = 27) | 92 ± 11 (n = 28) | 10.2 ± 0.7 (n = 21) | 9.8 ± 0.9 (n = 26) | 73.6 ± 4.4 (n = 16) | 1.62 ± 0.10 (n = 16) | 20.4 ± 2.7 (n = 20) |
| **Skinned trabeculae** | | Po | pCa50 | | **Actomyosin ATPase activity** | **Tension cost** | |
| | | mN mm$^{-2}$ | mN mm$^{-2}$ | | pmol*μl$^{-1}$*s$^{-1}$ | pmol*μl$^{-1}$*s$^{-1}$/mN mm$^{-2}$ | |
| WT (N = 3) | | 31 ± 4 (n = 9) | 5.93 ± 0.03 (n = 9) | | 685 ± 89.8 (n = 9) | 22.4 ± 1.12 (n = 9) | |
| *Obscn* KO (N = 3) | | 30 ± 2.3 (n = 7) | 5.91 ± 0.03 (n = 7) | | 583 ± 96.3 (n = 7) | 19.5 ± 2.87 (n = 7) | |

Data are means ± SE; N, number of mice in the group; n, number of myofibrils/skinned trabeculae; RT, resting tension; Po, maximal Ca$^{2+}$-activated tension; $D_{slow}$, slow-phase duration. In skinned trabeculae, pCa50 (50% of maximal Ca$^{2+}$-dependent tension), maximal (actomyosin) ATPase activity, and tension cost (actomyosin ATPase activity/tension) were measured. Statistical tests for all measurements: one-way ANOVA with Tukey's correction. *P < 0.05; **P < 0.01.

reduced in those from *Obscn* KO compared with those from WT mice at high frequencies (4 Hz and above). As a representative example, at 8 Hz, which corresponds to the physiological mouse heart rate, the peak tension (mN/mm²) was 14.0 ± 5.1 and 33.5 ± 6.5 in *Obscn* KO and WT trabeculae, respectively. The increase in contractile force in response to Iso was also significantly reduced in *Obscn* KO compared with WT trabeculae. Importantly, the β-adrenergic positive inotropic response was severely impaired at a high rate, with no contractile reserve present at 8 Hz. The time course of twitch contraction was preserved in *Obscn* KO at baseline. Also under Iso treatment, we observed that the peak contraction time was similarly shortened in *Obscn* KO and WT trabeculae at all frequencies (Fig. 3 C). However, the acceleration

of relaxation (i.e., lusitropic response) was impaired in *Obscn* KO trabeculae. Specifically, at 8 Hz, RT50% under Iso treatment was not significantly shorter compared with baseline (Fig. 3 D).

### SR uptake and SERCA function are impaired in the *Obscn* KO myocardium under β-adrenergic stimulation

The fraction of Ca$^{2+}$ that recirculates through the SR during subsequent beats (SR-Ca$^{2+}$ recirculation fraction, SR-RF) was estimated from the decay of contraction amplitude during dissipation of twitch potentiation induced by a period of rapid pacing rate (5 Hz). RF is represented by the slope of the linear relationship between the relative amplitude of subsequent contractions (Fn versus Fn + 1) during the decay of twitch

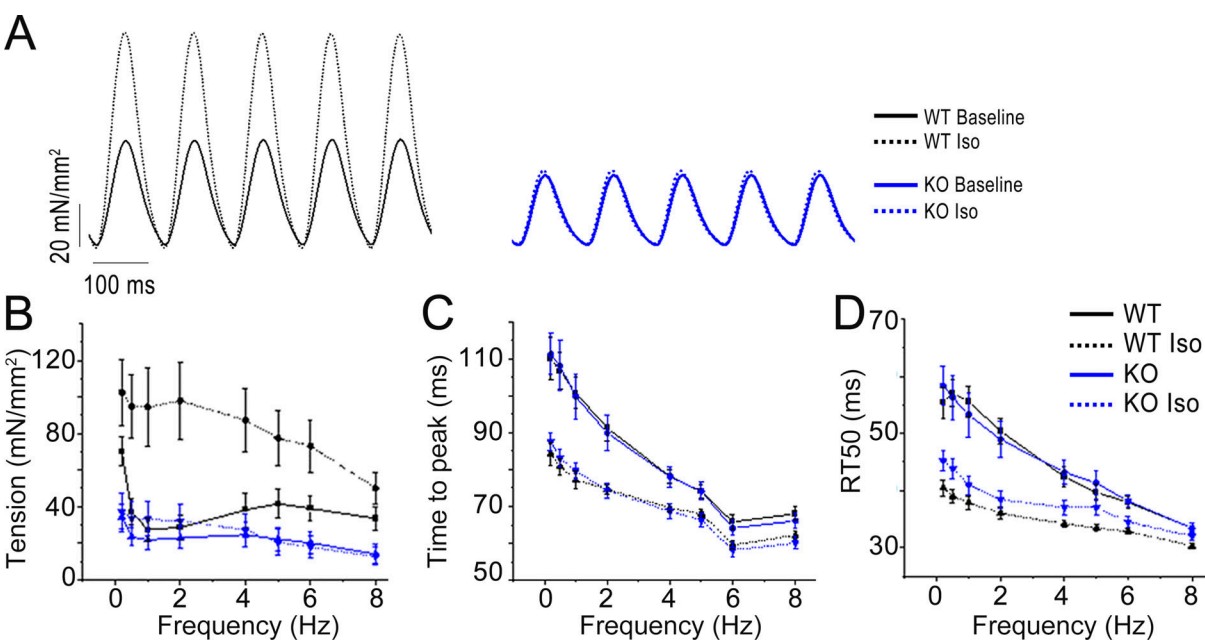

**Figure 3. Steady-state isometric twitches and short-term interval–force relationship. (A)** Representative force recordings from trabeculae of 8-mo-old WT and *Obscn* KO male mice, stimulated at 5 Hz, in the absence and in the presence of Iso 0.1 μM. **(B)** Relationship between active twitch force and stimulation frequency (0.1–8 Hz) in trabeculae from WT and *Obscn* KO mice, in the absence and in the presence of Iso 0.1 μM. **(C and D)** (C) Time to peak and (D) time from peak to 50% relaxation measured in steady-state twitches at different stimulation frequencies (0.1–8 Hz) in trabeculae from WT and *Obscn* KO mice, in the absence and in the presence of Iso 0.1 μM. Data are means ± SEM from N = 5 WT mice, n = 10 trabeculae; and N = 5 *Obscn* KO mice, n = 12 trabeculae. Statistical tests: one-way ANOVA with Tukey's correction with *0.05 > P > 0.01; **0.01 > P > 0.001.

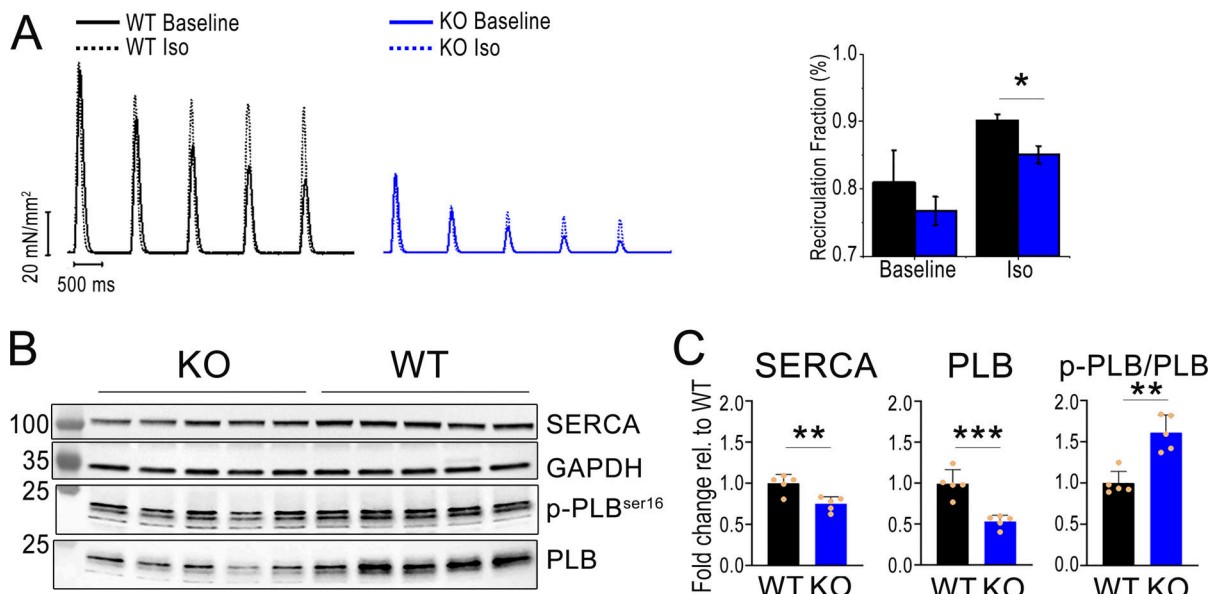

**Figure 4. Ca²⁺ recirculation fraction and SERCA/PLB expression. (A)** Representative Ca²⁺ recirculation fraction estimated by the decline of potentiated beats following a period of high stimulation rate in trabeculae from trabeculae of 8-mo-old WT and *Obscn* KO male mice, stimulated at 5 Hz, in the absence and in the presence of Iso 0.1 μm. **(B)** Representative western blots for total SERCA, PLB, phospho-PLB at serine 16, and GAPDH. **(C)** Protein expression levels of WT (*n* = 5) and *Obscn* KO (*n* = 5) hearts excised from 8-mo-old male mice. The intensity of individual bands was quantified following normalization to that of GAPDH. The mean relative intensity of WT samples was set at 1. **P < 0.01 and ***P < 0.001 versus WT, as determined by Student's *t* test. Source data are available for this figure: SourceData F4.

potentiation and reflects the relative contributions of SERCA versus the sodium–calcium exchanger in cytosolic Ca²⁺ removal (for the mouse cardiomyocytes, SR-RF approaches 85% in the absence of β-adrenergic–related SERCA activation). In *Obscn* KO trabeculae, the SR-RF was preserved at baseline, but significantly reduced under Iso, suggesting that in the heart of *Obscn* KO mice, SERCA function is preserved at baseline but cannot increase adequately under β-adrenergic stimulation (Fig. 4 A).

In western blot studies, we found that the total amount of SERCA protein was reduced in the *Obscn* KO myocardium. Moreover, the total amount of PLB protein was also decreased, whereas its phosphorylation level at Ser16 was increased (Fig. 4, B and C). Taken together, these results indicate that in the *Obscn* KO myocardium at baseline, a reduced inhibition of the SERCA pump by PLB (quantitatively lower and hyperphosphorylated) may functionally compensate for the reduced expression of the pump. At the same time, the increased tonic phosphorylation levels of PLB at the PKA site (Ser16) (Fig. 4 B), combined with the overall reduction in the total amount of both SERCA and PLB proteins, may explain the impaired increase in SERCA function under β-adrenergic stimulation (Iso).

### SR refractoriness is reduced and SR Ca²⁺ leakage increased in the *Obscn* KO myocardium

Mechanical restitution was studied by introducing a premature stimulus into a regular stimulus sequence: the associated contraction (extrasystole) was reduced in amplitude (Fig. 5 A). The amplitude of the extrasystolic beat increases as the interval preceding the premature stimulus increases, until the steady-state force is restored (mechanical restitution). Moreover, the subsequent beat (postextrasystolic beat) is greater than regular

beats as all Ca²⁺ sequestered in the previous two beats is released (post-postextrasystolic potentiation) (Cooper and Fry, 1990). Mechanical restitution curves show the fractional recovery of force (premature twitch force/steady-state twitch force) plotted against the premature stimulus interval. Mechanical restitution was faster in *Obscn* KO compared with WT trabeculae (Fig. 5 B). This result, in the absence of a faster SR Ca²⁺ uptake rate, suggests a shorter SR refractoriness in the *Obscn* KO myocardium, probably related to variations in RyR2 gating. β-adrenergic stimulation, by maximally accelerating SR Ca²⁺ uptake, accelerated mechanical restitution in both *Obscn* KO and WT preparations (Fig. 5 B) and abolished the difference in mechanical restitution between the two, likely indicating an increased RyR2 open probability in the *Obscn* KO trabeculae under Iso, which completely compensates for the reduced SR uptake. The post-postextrasystolic potentiation was not different between WT and *Obscn* KO trabeculae (Fig. 5 C). In line with the hypothesis that RyR2 open probability may be increased, thereby enhancing SR Ca²⁺ leak, in a subset of experiments we quantified the occurrence of small and large spontaneous Ca²⁺ oscillations (Fig. 5, C and D) in FluoForte-loaded cardiomyocytes superfused with Iso and subjected to an induction protocol (20-s high-frequency stimulation followed by 60-s stimulation pauses). In particular, small spontaneous Ca²⁺ oscillations have generally small amplitude compared with baseline (<30% of triggered Ca²⁺ transient) and slow rise velocity (three times below that of triggered Ca²⁺ transients). Large spontaneous Ca²⁺ oscillations have amplitude similar or higher than that of triggered Ca²⁺ transient and as rapid as triggered Ca²⁺ transients. In these experiments, FluoForte fluorescence was collected with a confocal microscope in a line scan. We found that the occurrence of spontaneous Ca²⁺

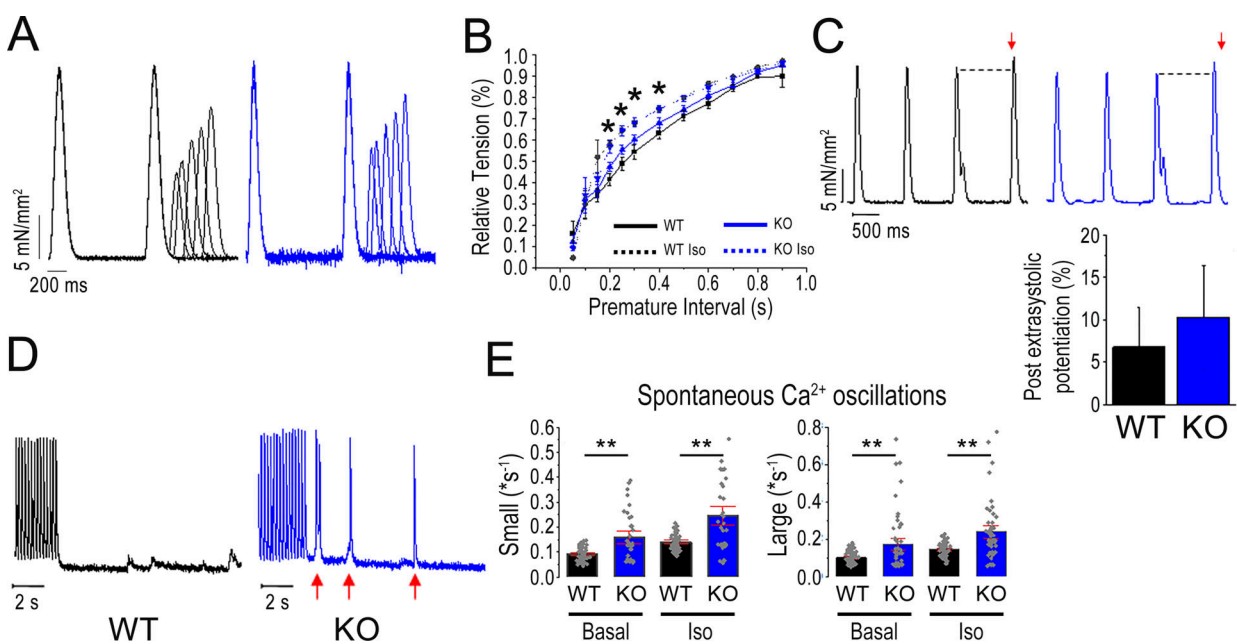

**Figure 5. Mechanical restitution and SR-Ca²⁺ leakage. (A)** Mechanical restitution protocol: representative traces from WT and *Obscn* KO trabeculae prepared from 8-mo-old male mice: steady-state stimulation, 1 Hz; premature interval, 250–600 ms. **(B)** Restitution curves show the fractional recovery of force (percentage of 1-Hz steady-state peak force) in response to the premature stimulus plotted against the premature interval. **(C)** Post-postextrasystolic potentiation is calculated by normalization of the postextrasystolic beat and the last regular beat following the extrasystole. **(D)** Representative traces showing the stimulation pause protocol used to elicit spontaneous Ca²⁺ events in WT and *Obscn* KO cardiomyocytes at basal conditions and in the presence of Iso 0.1 µmol/liter (Iso). Notably, *Obscn* KO cardiomyocytes showed frequent Ca²⁺ oscillations (red arrows). **(E)** Frequency of spontaneous Ca²⁺ waves and spontaneous Ca²⁺ transients during stimulation pauses in WT and *Obscn* KO cardiomyocytes at basal conditions and in the presence of 0.1 µmol/liter Iso. Data are the mean ± SEM from n = 55 WT (N = 5 mice), n = 35 *Obscn* KO (N = 5 mice) cardiomyocytes. Statistical tests: one-way ANOVA with Tukey's correction with *0.05 > P > 0.01.

oscillations was increased in *Obscn* KO cardiomyocytes compared with WT, both at baseline and under Iso.

### Ca²⁺ transient amplitude is reduced in *Obscn* KO cardiomyocytes

To further investigate the excitation–contraction coupling (E-CC) process, intracellular Ca²⁺ measurements were performed in Fura-2 AM–loaded cardiomyocytes during electrical field stimulation at 1–5 Hz, 35°C. Fura-2 AM fluorescence was collected in a wide field. Representative traces at 5 Hz are shown in Fig. 6 A and demonstrate that diastolic [Ca²⁺]ᵢ was increased and systolic [Ca²⁺]ᵢ was decreased in *Obscn* KO. Pacing frequency protocols in single cells were limited compared with intact muscles and cannot reproduce in vivo conditions, but the study of the frequency dependence of the two parameters (diastolic and systolic [Ca²⁺]ᵢ at 1, 3, and 5 Hz) allowed us to show that Ca²⁺ transient amplitude is always impaired, while the diastolic Ca²⁺ levels are normal at low pacing rates (Fig. 6 B). These results suggest that the structural alterations of the SR and TT observed in *Obscn* KO hearts (Fig. 2, F–H) may indeed have a functional impact on SR release/refractoriness.

### Absence of variations in action potential kinetics and conduction velocity but increased arrhythmogenic propensity in *Obscn* KO hearts

We optically mapped the transmembrane potential of cardiomyocytes to quantify action potential (AP) shape and propagation kinetics across the LV free wall in the isolated Langendorff-perfused hearts from *Obscn* KO and WT mice. To overdrive the sinus rhythm, APs were mapped, while the heart was paced at the apex at a stimulation frequency of 5 Hz (Fig. 7 A). We found that AP kinetics, specifically AP upstroke velocity, were unchanged (TTP in Fig. 7 B) even though slightly slower in the duration (APD₉₀, ms) in *Obscn* KO compared with WT hearts (Fig. 7 B). Additionally, we investigated conduction properties in the free wall and found no differences in local conduction velocity and wavefront regularity between *Obscn* KO and WT hearts (Fig. 7 C).

Next, we studied arrhythmia susceptibility following high-frequency burst pacing in the presence of the β-adrenergic agonist. Proarrhythmic effects observed in vivo are frequently not replicated in corresponding ex vivo preparations during mechanistic studies. To address this gap, a protocol has been developed for ex vivo electrophysiological pacing, employing 40-Hz bursts of 10 trials of 400 pulses each to induce arrhythmias. It is important to note that the protocol was not developed to serve as a physiological intervention. Rather, its purpose is to assess the proarrhythmic propensity in models that are inherently prone to arrhythmias. The protocol's objective is to emulate the Ca²⁺ overload that may occur under stress conditions. We found that *Obscn* KO hearts exhibited an increase in the occurrence of arrhythmias compared with WT hearts (Fig. 7 D). The increased SR Ca²⁺ leakage described above could be also related to this heightened arrhythmia susceptibility (Fig. 7 E).

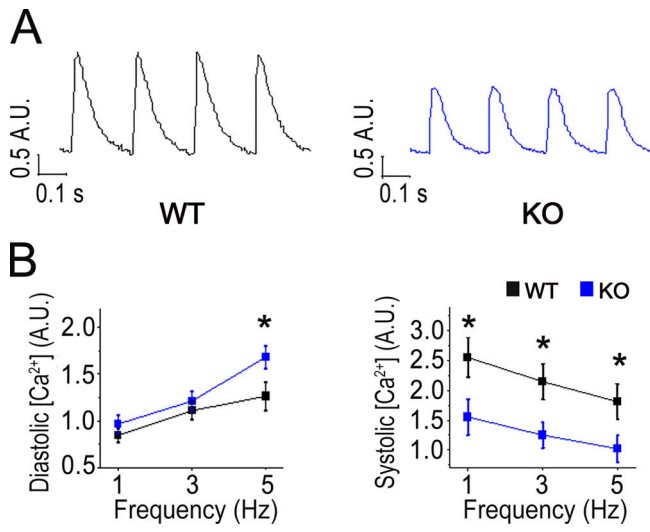

**Figure 6. Intracellular Ca²⁺ measurements in intact ventricular cardiomyocytes. (A)** Representative superimposed Ca²⁺ transients elicited at 5 Hz in WT and Obscn KO cardiomyocytes. **(B)** Mean ± SE of diastolic (left) and systolic (right) Ca²⁺ levels, expressed as arbitrary units of fluorescence intensity, during steady-state stimulation at different frequencies in $n$ = 55 WT cardiomyocytes ($N$ = 5 mice), $n$ = 35 Obscn KO cardiomyocytes ($N$ = 5 mice). *P < 0.05.

## Discussion

Here, we report results from in vivo and ex vivo studies aimed to characterize the effects of a global *Obscn* KO on cardiac functions. Echocardiography showed that *Obscn* KO hearts had significantly larger chamber volumes (+20% in end-diastolic and

end-systolic volumes), reduced fractional shortening, and impaired ejection fraction, consistent with DCM. These results are consistent with those of Fujita (Fujita et al., 2022, *Preprint*), which were limited to estimating diameters and fractional shortening in 2D measurements. By definition, DCM is characterized by dilation of the left or both ventricles with systolic dysfunction, which is not caused by ischemic or valvular heart disease. The hallmark pathophysiological feature of DCM is reduced myocardial contractility that may result from impaired myofilament and/or E-CC functions (Hasenfuss et al., 1992; Schultheiss et al., 2019). Reduced myocardial contractility can be acutely counteracted by increasing ventricular volume to maintain cardiac output through the Frank–Starling mechanism. Over time, the persistently increased LV volumes lead to thinning of the ventricular walls, promoting the dilated LV appearance that is observed in overt DCM. Indeed, in the *Obscn* KO mice, LV stroke volume and total cardiac output were preserved. In addition, LV wall thickness and dimensions were normal.

Can we therefore define the *Obscn* KO model as a DCM model?

The most suitable response to this question would be that this is a compensated form of DCM. In this model, impaired pump function (with evidence of pump failure) may only become apparent under certain conditions, such as physical exercise, aging, or secondary insults that may occur in addition to the genetic insult. This agrees with previous evidence in skeletal muscle tissue of *Obscn* KO mice (Randazzo et al., 2013, 2017a). The in vitro studies have contributed to clarify this hypothesis of a compensated form of DCM.

The investigation was conducted to determine the factors that contribute to diminished contractility in *Obscn* KO mice.

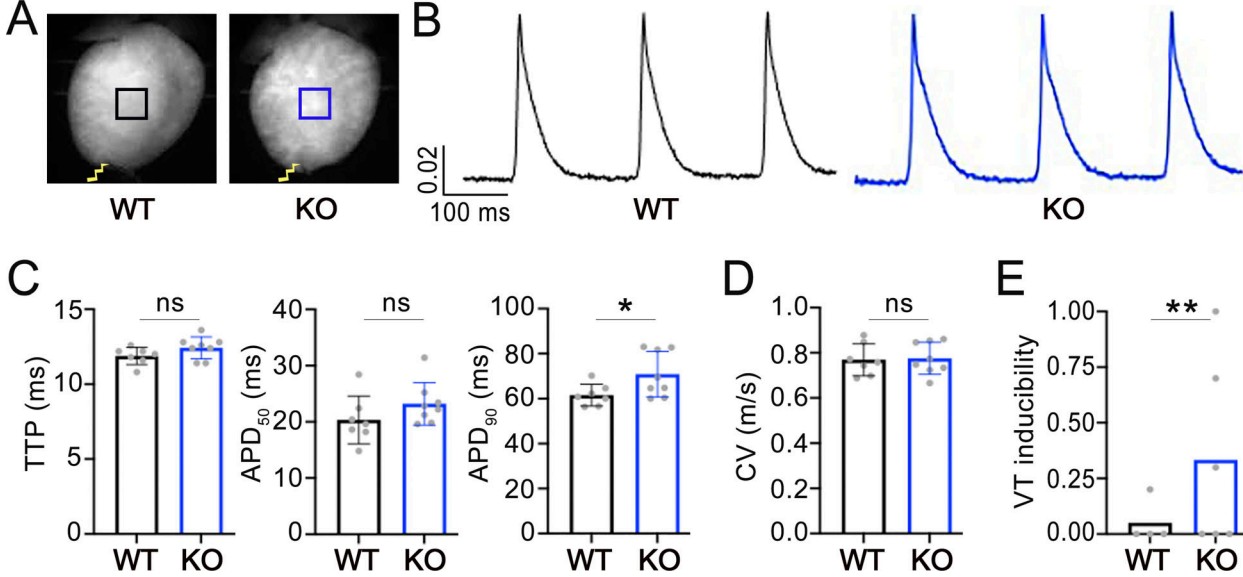

**Figure 7. AP kinetics, conduction, and arrhythmogenicity in the Langendorff-perfused *Obscn* KO mouse heart. (A)** Representative fluorescence images showing WT and *Obscn* KO mouse heart. Scale bar = 2 mm. The hearts were electrically paced at the apex (yellow bolt). **(B)** Fluorescent signals (ΔF/F) were extracted from the LV (black ROI: CTRL, blue ROI: *Obscn* KO) during a burst of stimuli at 5 Hz. **(C)** Average AP upstroke kinetics (TTP) and AP duration at 50% and 90% of repolarization (APD50 and APD90, respectively) in the LV free wall of WT (black) and *Obscn* KO (blue) hearts. **(D)** Average CV (m/s) in CTRL and *Obscn* KO LV free walls. **(E)** VT inducibility in WT and *Obscn* KO hearts. Data were collected from seven WT to eight *Obscn* KO mouse hearts and reported as the mean ± SEM. Student's *t* test analysis was performed with *0.05 > P > 0.01 and **0.01 > P > 0.001. CV, conduction velocity; TTP, time to peak; VT, ventricular tachycardia.

Sarcomere function was preserved, as ventricular myofibrils maintained normal force production, activation/relaxation kinetics, and Ca²⁺ sensitivity similar to WT. However, *Obscn* KO cardiomyocytes displayed reduced SR extension and swollen TT, although TT density was maintained. Indeed, intact muscle function was impaired, with reduced active tension at high frequencies, impaired β-adrenergic responses, and faster mechanical restitution, together with reduced SR refractoriness. Other findings included elevated diastolic Ca²⁺ levels, slower Ca²⁺ transient decay, and altered SERCA/PLB expression and phosphorylation levels. Although the duration of AP remained normal, we observed increased susceptibility to spontaneous Ca²⁺ oscillations and reentrant conduction under β-adrenergic stimulation, despite unchanged conduction velocity. These findings suggest abnormalities in the E-CC mechanisms.

In the next section, these alterations observed in vitro are discussed one by one and contextualized in support of the hypothesis of a compensated DCM phenotype as a consequence of the lack of obscurin.

### Preserved myofilament function

Variants in the most common myofilament DCM genes such as those expressing titin (Vikhorev et al., 2022), troponin T (Reda and Chandra, 2019), or troponin C (Dweck et al., 2008, 2010) directly cause variations in maximal force, passive tension, cross-bridge kinetics, as well as variations (often a reduction) in myofilament Ca²⁺ sensitivity and length-dependent activation. Interestingly, less common DCM variants in connecting proteins, such as dystrophin, which links the myofilament to the sarcolemma, or SR proteins (e.g., PLB, junctophilin), cause changes in myofilament function (e.g., reduced force, slow cross-bridge kinetics, changes in Ca²⁺ sensitivity) by activating intracellular pathways that target the myofilament with posttranslational modifications (phosphorylation, nitrosylation).

Unexpectedly, the *Obscn* KO model did not show any changes in the cross-bridge cycle or other sarcomere properties. To corroborate the results, mechanical measurements were performed on two different types of demembranated preparations, myofibrils and skinned trabeculae. In the latter case, the force measurements were also combined with energetic measurements allowing a direct assessment of the energetic cost of contraction. Notably, not only the active properties of the sarcomere were unchanged, but so were the passive properties. As described above, myofibrils were subjected to Ca²⁺ activation and relaxation protocols and no differences were observed between *Obscn* KO and WT mice in key parameters such as Po, RT, $k_{ACT}$ and $k_{TR}$, or tension relaxation kinetics. Sarcomere energetics was assessed by measuring both isometric tension and ATPase activity in demembranated ventricular trabeculae at different Ca²⁺ levels. Consistent with the myofibril data, no differences were found between *Obscn* KO and WT in maximal or RT and ATPase activity (both resting and Ca²⁺-activated). Myofilament Ca²⁺ sensitivity (pCa50) was also preserved (and not reduced as often observed in DCM, Chung et al. [2016]) in the *Obscn* KO skinned trabeculae, at least at our reference sarcomere length (SL) of 2.20 ± 0.05. In fact, the length-dependent activation (i.e., the reduction of pCa50% with increasing SL) was

not directly examined when assessing Ca²⁺ sensitivity, and this remains one of the limitations of our study to date. Consistent with our observation of preserved sarcomere function, (Fujita et al., 2022, *Preprint*) did not observe any structural changes in the organization and integrity of the sarcomeres in the *Obscn* KO mouse, either at the level of the A-band or at the level of the Z-disk.

### Reduced SR volume and twitch amplitude

Despite the absence of sarcomere changes, intact *Obscn* KO trabeculae exhibited significantly reduced active tension during high-frequency contractions that approximate the in vivo heart rate of mice 480–540 bpm (8–9 Hz), as well as during rested-state contractions (0.1–0.5 Hz). Rested-state contractions are widely accepted as a measure of the contractile reserve of the myocardium (Ferrantini et al., 2014), which was significantly reduced in *Obscn* KO trabeculae. This reduction in active tension suggests an underlying impairment in cardiomyocyte contractile function that may not be directly related to the morphofunctional integrity of the sarcomeres themselves. The reduced contractile force at both high and low frequencies (i.e., the loss of rate adaptation of twitch amplitude) raised important questions about the efficiency of the E-CC process. The intracellular Ca²⁺ measurements in cardiomyocytes clearly showed that diastolic intracellular Ca²⁺ concentration ([Ca²⁺]ᵢ) was increased, whereas peak systolic [Ca²⁺]ᵢ was decreased in *Obscn* KO cardiomyocytes. Although pacing frequency protocols applied to single cells have limitations compared with experiments performed on intact muscle tissue, we found that the amplitude of Ca²⁺ transients is consistently impaired at 5 Hz, consistent with a significant impairment of force production at high pacing rates. In this context, it is noteworthy to mention that the absence of obscurin is sufficient to reduce SR extension and that there is an evident alteration of TT morphology in *Obscn* KO cardiomyocytes. The decrease in SR volume is consistent with the reduction in Ca²⁺ transient amplitude and the decrease in contractile reserve (Orchard and Brette, 2008; Setterberg et al., 2021). However, SR collapse coupled with an increased area of TT may also suggest that in *Obscn* KO, these morphological changes may enhance plasmalemmal Ca²⁺ flux to compensate for the inefficiency of SR Ca²⁺ release (Protasi et al., 2023). Consistent with this interpretation, we observed that post-extrasystolic potentiation after the early premature extra beats (known to reflect the amplitude of the Ca²⁺ current) was not simply preserved (suggestive of normal TT density and preserved Ca²⁺ current) but tended to be increased in *Obscn* KO (an indication that Ca²⁺ current could be even be enhanced, Fig. 5 C) (Sprenkeler and Vos, 2016). Reduced twitch amplitude is indeed more evident under inotropic interventions (high frequency and Iso). These results also indicate that cardiomyocytes possess a remarkable capacity to remodel TT in response to a decrease in SR volume, as previously documented (Swift et al., 2012).

The observed reduction in twitch and Ca²⁺ transient amplitudes and the reduced contractile reserve raise the question of whether our *Obscn* KO model can be defined as a model of reduced myocardial contractility. This differs from the in vivo hemodynamic data evaluating the max dP/dt, which was found

to be completely normal in Fujita et al. (2022), *Preprint*. Filling volume adjustments may reconcile the in vitro and in vivo data. Indeed, as largely discussed above, we find that mice in the *Obscn* KO model have increased working volumes.

### SR collapse and impaired β-adrenergic response

Dysregulation of the β-adrenergic response may be a key pathogenic determinant in *Obscn* KO mice. The evidence we have collected relates to twitch contraction measurements performed at baseline and during Iso stimulation to simulate β-adrenergic activation. The increase in contractile force in response to Iso was significantly reduced in *Obscn* KO compared with WT samples. In particular, the β-adrenergic positive inotropic response was markedly impaired at high frequencies, with no contractile reserve at 8 Hz. The time course of twitch contractions at baseline was preserved in *Obscn* KO. Under Iso (compared with baseline), the acceleration of relaxation was impaired in *Obscn* KO. In particular, at 8 Hz, the RT50% under Iso did not show a significant reduction compared with baseline. This means that the rate of relaxation does change in both models, but to a lesser degree in the *Obscn* KO. This evidence supports an impairment of the β-adrenergic response, which is consistent with the loss of SR integrity. Indeed, in *Obscn* KO hearts, ultrastructural SR collapse and volume reduction were confirmed by western blot analysis, which showed a decrease in the total SERCA protein levels in the *Obscn* KO myocardium (Fig. 3, B and C). Additionally, the overall amount of the PLB protein was reduced, while its phosphorylation at Ser16 was elevated (Fig. 4 C). These findings suggest that in the *Obscn* KO myocardium under baseline conditions, the diminished inhibition of SERCA by PLB, due to its lower levels and increased phosphorylation, may compensate functionally for the reduction of both SR volume and expression levels of the pump. However, the increased baseline phosphorylation of PLB at the PKA site (Ser16), along with the overall reduction in SERCA and PLB proteins, could be an additional explanation for the impaired enhancement of SERCA activity during β-adrenergic stimulation (i.e., Iso). These data suggest that the observed multifunction of the β-adrenergic response in *Obscn* KO mice relies on the loss of SR volume and, as a consequence, on the dysregulation of the intracellular pathway rather than (or in addition to) the receptor expression levels itself or other targets of PKA-mediated phosphorylation (cTnI or cMyBP-C). This point differs from the data reported by Fujita et al. (2022), *Preprint*, because in that study, the *Obscn* KO model shows a perfectly adequate response to β-adrenergic stimulation (with dopamine) in vivo. However, it should be noted that our experiments were performed on denervated hearts, and thus, the lack of an adequate positive inotropic response observed ex vivo could be compensated in vivo by a different adrenergic tone or an increased density of sympathetic terminals. A limitation of the current study is the lack of measurement of β-receptor density in the myocardium of our mouse models. An additional contributor to the blunted positive limb of the force–frequency response could be the instability of the residual SR volume and its additional impairment under β-adrenergic stress, possibly due to RyR2 hyperphosphorylation by PKA.

### SR instability and arrhythmias

In *Obscn* KO mice, the arrhythmogenic propensity was assessed at both the organ and the single-cell level. In the Langendorff-perfused hearts, we observed an increased susceptibility to re-entrant conduction, in the absence of variations in membrane electrophysiology (e.g., AP duration) and conduction velocity. In trabeculae, spontaneous activity occurred in >25% of the *Obscn* KO preparations, whereas premature contractions occurred in <8% of the controls. In *Obscn* KO cardiomyocytes, we found increased diastolic $Ca^{2+}$ levels and an increased occurrence of spontaneous $Ca^{2+}$ oscillations, i.e., cellular triggers for arrhythmia initiation, at baseline and even more during β-adrenergic stimulation. Taken together, the data show that the occurrence of cellular triggers is increased and becomes sufficient to initiate reentrant conduction, even in the absence of specific substrates at the tissue level (no conduction velocity variation and no evidence of interstitial fibrosis) (Fig. 2 C).

It remains to be explained why *Obscn* KO cardiomyocytes exhibit more spontaneous SR $Ca^{2+}$ release than controls. The data collected suggest at least three concurrent mechanisms. One can be simply related to the sustained elevation of diastolic $Ca^{2+}$ (Fig. 5) in *Obscn* KO cardiomyocytes. Secondly, prolonged exposure to elevated intracellular $Ca^{2+}$ levels could lead, either directly or through activation by CaMKII, to an increased probability of ryanodine receptor (RyR2) opening and, consequently, to increased spontaneous SR $Ca^{2+}$ release. A third contribution is related to the increased baseline phosphorylation of PLB at the PKA site, which accelerates SERCA activity and thus SR refilling in *Obscn* KO cardiomyocytes, This, in turn, exerts an influence on RyR2 recovery through luminal control. The mechanical restitution data align with this interpretation.

### Conclusions

Our echocardiographic measurements show that the two key features of the human DCM phenotype (LV dilation and hypocontractility) are replicated in the *Obscn* KO mouse model. However, the decrease in contractility is mild and is compensated by an increase in EDV, resulting in a compensated DCM phenotype. In vitro, we found that despite the complete loss of obscurin, sarcomere function is unexpectedly preserved, but twitch amplitude and positive inotropic responses are impaired due to a dramatic SR collapse and volume reduction. In addition, the risk of arrhythmias is increased in the *Obscn* KO mouse model, with no major tissue substrates (no variations in conduction velocity, no fibrosis). We highlight the dysregulation of the β-adrenergic response, and the reduced SR refractoriness associated with SR $Ca^{2+}$ leak as key mechanistic points underlying the reduced SR volume (contractile reserve) and increased arrhythmogenicity. A potential limitation of the study is that only mature adult mice were investigated (8 mo of age). However, it is reasonable to hypothesize that protracted aging or other stress factors (physical exercise; pregnancy) might be involved in developing a classic DCM phenotype.

### Data availability

No new data were generated or analyzed in support of this study.

## Acknowledgments

Henk L. Granzier served as editor.

We thank Li Cui for the help with preparing and capturing the electron micrographs.

This work was supported by grants to V. Sorrentino from the Center for Gene Therapy and Drugs Based on RNA Technology, funded in the framework of the National Recovery and Resilience Plan (NRRP), M4C2 Inv, 1.4 CUP B63C22000610006-Spoke 1, and the Tuscan Health Ecosystem, funded in the framework of the National Recovery and Resilience Plan (NRRP), Missione 4 Componente 2 Inv. 1.5 CUP B63C22000680007-Spoke 7. Both grants are, in turn, funded by the European Union-Next Generation EU, and by grants from the National Institutes of Health (HL152251, HL128457), Novo Nordisk Foundation (NNF22OC0079368), Aarhus University Research Foundation (AUFF-E-2022-7-9), Lundbeck Foundation (R396-2022-189), and Independent Research Fund Denmark (DFF, #3165-00028B) to S. Lange.

Author contributions: J.M. Pioner: conceptualization, data curation, formal analysis, investigation, methodology, project administration, resources, supervision, validation, visualization, and writing—original draft, review, and editing. E. Pierantozzi: conceptualization, formal analysis, investigation, resources, validation, visualization, and writing—original draft, review, and editing. R. Coppini: conceptualization, data curation, formal analysis, investigation, and visualization. E.M. Rubino: formal analysis, investigation, and resources. V. Biasci: investigation. G. Vitale: investigation. A. Laurino: formal analysis, investigation, validation, and writing—original draft. L. Santini: data curation, formal analysis, investigation, and validation. M. Scardigli: data curation, investigation, and writing—original draft, review, and editing. R. Davide: investigation and writing—review and editing. C. Olianti: conceptualization, investigation, methodology, validation, and writing—review and editing. M. Serano: formal analysis, investigation, and resources. D. Rossi: validation. C. Tesi: conceptualization, methodology, supervision, validation, and writing—review and editing. E. Cerbai: supervision. S. Lange: formal analysis, investigation, resources, visualization, and writing—review and editing. C. Reggiani: conceptualization, visualization, and writing—original draft, review, and editing. L. Sacconi: formal analysis, investigation, and writing—review and editing. C. Poggesi: conceptualization, funding acquisition, methodology, project administration, supervision, and writing—original draft, review, and editing. C. Ferrantini: conceptualization, data curation, formal analysis, funding acquisition, investigation, methodology, project administration, resources, software, supervision, validation, visualization, and writing—original draft, review, and editing. V. Sorrentino: conceptualization, data curation, funding acquisition, project administration, resources, supervision, validation, visualization, and writing—original draft, review, and editing.

Disclosures: The authors declare no competing interests exist.

Submitted: 13 October 2024

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
