## [Peer Review File · The Journal of General Physiology]

Obscurin deficiency leads to compensated dilated cardiomyopathy and increased arrhythmias

Jose Manuel Pioner, Enrico PIERANTOZZI, Raffaele Coppini, Egidio Rubino, Valentina Biasci, Giulia Vitale, Annunziata Laurino, Lorenzo Santini, Marina Scardigli, Davide Randazzo, Camilla Olianti, Matteo Serano, Daniela Rossi, Chiara Tesi, Elisabetta Cerbai, Stephan Lange, Carlo Reggiani, Leonardo Sacconi, Corrado Poggesi, Cecilia Ferrantini, and Vincenzo Sorrentino

Corresponding Author(s): Vincenzo Sorrentino, University of Siena

Review Timeline:

Submission Date:	October 13, 2024
Editorial Decision:	November 23, 2024
Revision Received:	March 17, 2025
Editorial Decision:	April 3, 2025
Revision Received:	April 15, 2025

Editor: Henk Granzier

Transaction Report:

DOI: <https://doi.org/10.1085/jgp.202413696>

November 25, 2024

Prof. Vincenzo Sorrentino
University of Siena
Molecular and Developmental Medicine
via Aldo Moro 2
Siena 53100
Italy

Re: 202413696

Dear Prof. Sorrentino,

Thank you for submitting your manuscript, entitled "Obscurin deficiency leads to compensated dilated cardiomyopathy with reduced inotropic reserve and increased arrhythmias" to JGP. Your manuscript has now been seen by 3 reviewers, whose comments are appended below. You will see that the reviewers were enthusiastic about the study and its potential impact but raised some concerns that should be addressed before further consideration of the manuscript at JGP. We also request that you provide exact p-values in all figures. Additionally, in Fig 7C (right-hand panel), the p-value is 0.0449, but the text states that there was no significant change in APD. Please correct.

We hope that you will be able to submit a revised manuscript that addresses these points, which we believe will pose no problems, and which may be re-reviewed. In addition, please do not hesitate to contact me (via the editorial office) if you feel that a discussion of the reviewers' and editors' comments would be helpful.

Please submit your revised manuscript via the link below, along with a point-by-point letter that details your response to the reviewers' and editors' comments, as well as a copy of the text with alterations highlighted (boldfaced or underlined). If the article is eventually accepted, it would include a 'revised date' as well as submitted and accepted dates. If we do not receive the revised manuscript within one year, we will regard the article as having been withdrawn. We would be willing to receive a revision of the manuscript at a later time, but the manuscript will then be treated as a new submission, with a new manuscript number.

Please pay particular attention to recent changes to our instructions to authors in the following sections: Data presentation, Blinding and randomization and Statistical analysis, under Materials and Methods, as shown here: <https://rupress.org/jgp/pages/submission-guidelines#prepare>. Re-review will be contingent on inclusion of the required information (including for data added during revision) and demonstration of the experimental reproducibility of the results. Also, To improve the reproducibility of published content, we have partnered with SciScore. Authors are prompted in eJP to copy and paste the Materials and Methods section of their manuscript for a SciScore assessment when submitting their revised manuscript. Authors are encouraged (not required) to further revise their Materials and Methods if the SciScore is below 4. More information can be found here: <https://rupress.org/jgp/pages/submission-guidelines#sciscore>.

Please note, JGP now requires authors to submit Source Data used to generate figures containing gels and Western blots with all revised manuscripts (when applicable). This Source Data consists of fully uncropped and unprocessed images for each gel/blot displayed in the main and supplemental figures. If your paper includes cropped gel and/or blot images, please be sure to provide one Source Data file for each figure that contains gels and/or blots along with your revised manuscript files. File names for Source Data figures should be alphanumeric without any spaces or special characters (i.e., SourceDataF#, where F# refers to the associated main figure number or SourceDataFS# for those associated with Supplementary figures). The lanes of the gels/blots should be labeled as they are in the associated figure, the place where cropping was applied should be marked (with a box), and molecular weight/size standards should be labeled wherever possible. Source Data files will be made available to reviewers during evaluation of revised manuscripts and, if your paper is eventually published in JGP, the files will be directly linked to specific figures in the published article.

Source Data Figures should be provided as individual PDF files (one file per figure). Authors should endeavor to retain a minimum resolution of 300 dpi or pixels per inch. Please review our instructions for export from Photoshop, Illustrator, and PowerPoint here: <https://rupress.org/jgp/pages/submission-guidelines#revised>

Whilst you are revising your manuscript, we ask that you consider whether you have any artwork that might be suitable for the cover of JGP. Microscopy images are particularly good for cover artwork, but other types of image can be very effective, so we encourage you to be creative. Please don't restrict yourself to images from the paper; an image that is relevant to the work described would be just as suitable. Images should be a minimum resolution of 300 dpi. To see recent examples, visit the following page and click on 'Show covers? Yes': <https://jgp.rupress.org/content/by/year>

Thank you for submitting your interesting research to JGP.

Please submit your revised manuscript, and any associated files, via this link:
Link Not Available

Sincerely,

Henk L. Granzier, Ph.D.
On behalf of Journal of General Physiology

Journal of General Physiology's mission is to publish mechanistic and quantitative molecular and cellular physiology of the highest quality; to provide a best-in-class author experience; and to nurture future generations of independent researchers.

Reviewer #1 (Comments to the Authors):

"Obscurin deficiency leads to compensated dilated cardiomyopathy with reduced inotropic reserve and increased arrhythmias", submitted by Pioner et al., reports on the dilated cardiomyopathy present in mice lacking obscurin (Obscn KO). The experiments are comprehensive and wide ranging, with both in vivo and in vitro studies, with the latter including both intact and skinned trabecular preparations. The results, which are extensive, are at points difficult to follow, but the overall conclusion, that the KO mice show evidence of altered ability to take up Ca²⁺ into the SR, and of increased Ca²⁺ leak through RyR2 from the SR into the myoplasm, seem well supported. The paper also presents an excellent review of the literature in the introduction. It would be strengthened by consideration of the following points.

Major:

1. Abstract: The abstract is confusing. It should be rewritten so that it clearly states the results in simple declarative sentences and ends with the overall conclusions the authors draw.

2. Methods: What membrane-selective dyes were used to stain t-tubules? What were the conditions used? What were the sources of the primary antibodies used in Western blots? Experimental details (dilutions, incubation times, etc.) are also needed.

What is the physiological relevance of using "40 Hz bursts (10 trials of 400 pulses each)" to induce arrhythmias?

3. Fig. 2: Panel C is barely visible and is not described in the figure legend.

4. Fig. 4: It is difficult to understand from the western blots where the values in the bar graphs for SERCA, PLB and phosphorylated PLB come from. The graphs show dots that do not seem to correlate with the blots to the left. Furthermore, the variability, especially in SERCA, in blots of both WT and KO is concerning.

5. Fig. 5: The text refers to "spontaneous Ca²⁺ waves and spontaneous Ca²⁺ transients during stimulation pauses", but the data only seem to illustrate the latter. What is the frequency in otherwise resting myocytes?

In panel B, the results seem to indicate that isoproterenol allows the KO to compensate completely (not "to some extent") for the reduced SR uptake rates.

6. Fig. 6: Wouldn't diastolic Ca²⁺ levels in KO be expected to be elevated even at low pacing rates, if RyR2 leak is involved? Or is the myopathy associated with an increased sensitivity of the RyR2 to Ca²⁺ but only when the levels are increased by stimulation?

It seems doubtful that the method in Panel C is sufficient to conclude that the morphology of the t-tubule system is normal. What do the green and red in Fig. 6C signify? The images seem quite different, with many more and larger discontinuities in the t-system in the KO than in WT. If so, could these help explain some of the results?

7. The SR in KO skeletal muscle is reduced compared to WT. Why was it not examined here? A reduced SR compartment would explain many of the results related to Ca²⁺ uptake, ATPase activity, recycling, etc.,

8. Discussion: The discussion of DCM did not consider the possibility that, had the mice been aged further, they would have developed a classic DCM phenotype.

The authors mention that there is no difference in Ca²⁺-ATPase, but the data in Table 1 indicate that there is a difference of ~15%. This may not be statistically significant, but given the variability (e.g., Fig. 4), this is not surprising. In fact, as just noted, a

difference in enzyme activity could account for many of the results here.

".., likely due to a higher SR Ca²⁺ uptake rate" It seems contradictory to argue that the Ca²⁺ uptake rate is elevated when SERCA levels and ATPase activities are reduced, diastolic Ca²⁺ levels are high, etc.

The discussion might include a more integrated consideration of the interplay between the changes in Ca²⁺ storage and Ca²⁺ leak that they observe.

Minor:

1. The English could use some editing for typographical errors, word choice (e.g., better terms for "layout", "set up", "dilatative", "heighted", "morpho-functional", etc.) and brevity.
2. Order of references: The authors might consider putting these in alphabetical order or by year of publication. As is, the order seems random.

Reviewer #2 (Comments to the Authors):

This manuscript characterises the cardiac phenotype of Obscurin knockout mice. The mice were originally published in 2009, but that study focused on skeletal muscle, so it is good to see that finally the heart is studied more closely. The careful analysis shows that while the mice do not have much of a phenotype at baseline, they lack contractile reserve upon stress, e.g. beta adrenergic response is completely absent. The authors then go on to perform a thorough characterisation of the ECC phenotype and find Ca leakage and some dysregulation of SERCA and phospholamban expression and phosphorylation.

The work is a careful analysis of the functional phenotype of Obscurin knockout hearts and shows that while there is nothing massively dramatic going on, these mice might be poised to respond with cardiac complications in a stress setting (physical exercise; pregnancy; age). From the Material & Methods it appears that all the studies were done in 2 months old animals; is this really the case? That strikes me as a bit young, since with mouse models phenotypes remain hidden for months; do the authors have any insights how ageing affects the animals? Along a similar line, only once it is mentioned that male hearts were used. Many mouse studies have now shown differences in the response between males and females (which is even more apparent in human patients); can the authors confirm that all the investigations were done in males, or tease apart data for males and females if possible? Also mention in the figure legend (e.g. Figure 1) which sex was used? Both these aspects do not necessarily warrant new experiments in this instance, but should be expanded on in the manuscript.

I am also wondering whether there is additional electron microscopy data available on the heart. The Lange et al., 2009 paper only shows zoom ins on individual skeletal muscle sarcomeres and since the authors claim here that they see a compensated DCM phenotype, it would be really interesting to see whether there are less distinct M-bands and more convoluted intercalated discs in these hearts, cellular phenotypes of full blown DCM seen by others (e.g. Schoenauer et al., 2011; PMID 21069531; Ehler et al., 2001; PMID 11352937).

Otherwise there are just minor errors that need addressing:

1. There is a glitch with the citations (three authors cited in the text and the like), which needs to be fixed.
2. Abstract: Obscurin is -a- giant sarcomeric
3. page 3: signaling domains in obscurin: are at the C-terminus not at the N-terminus as stated here
4. page 5: Nouredine and Gehmlich - then there is a gap - something missing here?
5. page 6: dilated not dilatative
6. Figure 2C: shows what? Information missing from the figure legend; first time it is mentioned is in the Discussion...
7. The differences in SERCA and Plb expression look very noisy in the immunoblots; much noisier than the data in the quantification; again I am wondering whether a male/female difference might be at the root of this?
8. Figure 6: Actually the T-tubules do look a bit more purple in the Obscurin KO cardiomyocytes, so there might be a phenotype that is not picked up by the analysis they performed; ImageJ does not solve every problem. Try the algorithm that Candasamy et al., 2014; PMID 24280220; used for their analysis?
9. page 21: In this context it is noteworthy that - gap- observed - who?
10. page 22: line 2 from bottom: Obscn KO not OBSC KO (assuming we are still talking about mice...)
11. The discussion could do with some trimming; it is a bit long at the moment.

Reviewer #3 (Comments to the Authors):

The authors use a obscurin KO mouse model to investigate the role of this protein in cardiac function. They show that dilation of the heart takes place with reduced contractility. This is not due to myofilament dysfunction, but rather changes in calcium handling. Also a notable blunting of beta1-adrenergic receptor stimulation is seen.

I think that the paper is of interest and that most of the experiments are performed to a high standard. I do have a number of questions and comments that I hope would improve the paper.

1) The lack of increase in force after isoprenaline in the KO mice is intriguing. I think this is one of the key findings of the paper. Unfortunately the mechanism of this is not explored as elaborately as I would have liked it to be. All follow-up experiments do not look at the effects on PLB phosphorylation or Ca-transient kinetics do not study the effect after ISO. I think this is key to understanding the mechanism. Is beta1-AR signaling still intact? Can it phosphorylate cTnl or cMyBP-C but not PLB?

2) The Western blot data of SERCA and PLB in figure 4 is not of sufficient quality to ensure reliable quantification. PLB seems missing in most samples (even in WT), SERCA expression is highly variable. This has to be improved, because the interpretation of the results depend on it. The results showing higher PLB-Ser16 phosphorylation is entirely driven by a low PLB level, not a high Ser16 phosphorylation.

3) In Figure 3 you show that forces do not change, but that time to peak and time to baseline do change with ISO in the KO mice. Does that mean that velocities do change in both models, but to a lesser degree in the KO?

4) Figure 2C is not described in the paper nor the legend. There is also no quantification provided

5) The intro is well written but too long

Minor:

1) The symbols in figure 3B do not match the legend for the KO group.

2) In the text it is stated that the mice have a heart rate of 8-9 Hz. This is typically presented as bpm in in vivo experiments.

3) On page 7 you describe your animals as transgenic. These are not transgenic, they are knock-out animals. Also in that section you say heighted instead of weighed.

UNIVERSITÀ
DI SIENA
1240

DIPARTIMENTO DI MEDICINA MOLECOLARE
E DELLO SVILUPPO

March 17, 2025

Prof. Henk L. Granzier,
Associate Editor, Journal General Physiology

Dear Henk,

Please find enclosed a revised version of our manuscript entitled “Obscurin Deficiency Leads to Compensated Dilated Cardiomyopathy with Reduced Inotropic Reserve and Increased Arrhythmias” for consideration for publication in the Journal of General Physiology's “Special Issue on Contractile Systems, Mechanobiology, and Regulation.”

We appreciated the positive comments and the suggestions from all reviewers and have modified the manuscript accordingly. A detailed point-by-point response to reviewers' comments is provided in the following pages. All changes are reported with the Word track changes in the text. The positions of these changes are indicated in the response to reviewers' comments.

We believe that the revisions made have substantially improved the clarity and significance of the manuscript. We hope that you will now find the manuscript suitable for publication in the *Journal of General Physiology*.

Sincerely

Vincenzo Sorrentino

Reviewer #1 (Comments to the Authors):

"Obscurin deficiency leads to compensated dilated cardiomyopathy with reduced inotropic reserve and increased arrhythmias", submitted by Pioner et al., reports on the dilated cardiomyopathy present in mice lacking obscurin (Obscn KO). The experiments are comprehensive and wide ranging, with both in vivo and in vitro studies, with the latter including both intact and skinned trabecular preparations. The results, which are extensive, are at points difficult to follow, but the overall conclusion, that the KO mice show evidence of altered ability to take up Ca²⁺ into the SR, and of increased Ca²⁺ leak through RyR2 from the SR into the myoplasm, seem well supported. The paper also presents an excellent review of the literature in the introduction. It would be strengthened by consideration of the following points.

Major:

1. Abstract: The abstract is confusing. It should be rewritten so that it clearly states the results in simple declarative sentences and ends with the overall conclusions the authors draw.

The abstract has been completely revised according to the suggestions provided (**page 2, lines 1-22**).

2. Methods: What membrane-selective dyes were used to stain t-tubules? What were the conditions used?

For T-tubules architecture analysis, cardiomyocytes are stained by adding to the cell suspension 2 µg/mL of the voltage sensitive dye (VSD) Di-3-ANEPPDHQ (dissolved in ethanol). After washing, cells are resuspended in fresh Ca²⁺ free solution containing 10 µM blebbistatin, 4 µM cytochalasin D. Loaded preparations are used for experiments within 30 min. The staining and imaging sessions are performed at room temperature (20 °C) as previously reported (Crocini et al. 2016). We added this extended description to the methods section (**page 6, lines 24-35**).

What were the sources of the primary antibodies used in Western blots? Experimental details (dilutions, incubation times, etc.) are also needed.

The description of the methods related to the western blot experiments has been revised (**page 7, lines 35-45, and page 8, lines 1-13**) in response to the requests of Reviewer 1 and to incorporate the new experiments performed (see point 4). An additional reference (Mountian et al., 2001) has been added to the 'References' section.

What is the physiological relevance of using "40 Hz bursts (10 trials of 400 pulses each)" to induce arrhythmias?

Proarrhythmic effects observed in vivo are frequently not replicated in corresponding ex vivo preparations during mechanistic studies. To address this gap, a protocol has been developed for ex vivo electrophysiological pacing as previously reported (Scardigli M. et al. 2020), employing 40-Hz bursts of 10 trials of 400 pulses each to induce arrhythmias. It is important to note that the protocol was not developed to serve as a physiological intervention. Rather, its purpose is to assess the proarrhythmic propensity in models that are inherently prone to arrhythmias. The protocol's objective is to emulate the calcium overload that may occur under stress conditions.

3. Fig. 2: Panel C is barely visible and is not described in the figure legend.

The reviewer is correct; however, it should be noted that, on average, the amount of type III collagen present in a healthy heart is approximately 10% of the total collagen, which is primarily composed of type I

collagen (Horn MA, Trafford AW. doi: 10.1016/j.yjmcc.2015.11.005). The acquisition parameters used to detect the binding of the primary anti-Coll III antibody (ab7778) in this set of experiments are the same as those used in our previous study on skeletal muscle fibers from *Obscn* KO animals at rest and after intense physical activity (Randazzo et al. 2017). In those experiments, it was observed that, in fibers at rest (and thus in the absence of interstitial fibrosis or remodeling), the signal for type III collagen was "barely detectable." In contrast, fibers after intense exertion showed a much stronger and more widespread signal, indicating not only its biological significance but also that the antibody can detect changes in type III collagen expression. However, in the revised Panel C of Fig. 2, we have slightly adjusted the contrast and brightness of both immunofluorescence images to enhance the visibility of the inherently faint signal. These parameters were modified in the same way for both images. This adjustment does not alter the outcome of the analysis, which shows that no changes in type III collagen expression were observed between WT and KO hearts. The legend for the description of Panel C has now been added (**page 25, lines 15-29**). We apologize for the oversight. The manuscript has been revised accordingly (**page 9, lines 29-30**).

4. Fig. 4: It is difficult to understand from the western blots where the values in the bar graphs for SERCA, PLB and phosphorylated PLB come from. The graphs show dots that do not seem to correlate with the blots to the left. Furthermore, the variability, especially in SERCA, in blots of both WT and KO is concerning.

The reviewer is absolutely correct that the original figure chosen to represent our results did not clearly correlate with the quantitative analysis of the blots, as also noted by Reviewers 2 and 3. Accordingly, we performed new western blot experiments on total protein lysates prepared from 5 WT hearts and 5 KO hearts, taken from 8-month-old mice. The results obtained confirmed the data reported in the original submission regarding SERCA and PLB expression. Since the quality of the new blots is superior to the previous ones, panels B and C of Figure 4 now present these new experiments. The corresponding legend and Materials and Methods have been revised accordingly (**page 25, lines 42-48 and page 27, lines 1-2; page 7, lines 35-45, and page 8, lines 1-13**), also in line with previous suggestions (see point 2).

5. Fig. 5: The text refers to "spontaneous Ca²⁺ waves and spontaneous Ca²⁺ transients during stimulation pauses", but the data only seem to illustrate the latter. What is the frequency in otherwise resting myocytes?

We thank the reviewer for raising this point on the terminology. It is true that spontaneous Ca²⁺ waves and spontaneous Ca²⁺ transients refer to the observation of spontaneous calcium oscillations during diastole, often evaluated through the use of advanced resolution microscopy techniques, such as confocal microscopy.

In particular, spontaneous Ca²⁺ waves are Ca²⁺ oscillations with small amplitude compared to baseline generally (<30% of triggered calcium transient) and slow (rise velocity is generally 3 times below that of triggered calcium transients). Spontaneous Ca²⁺ transients are Ca²⁺ oscillations that have amplitude similar or higher than that of triggered calcium transient, and rapid as triggered calcium transients.

To clarify the presence of arrhythmic events observed in isolated cardiomyocytes from both WT and *Obscn* KO mouse hearts during resting intervals we replaced spontaneous Ca²⁺ waves with "small spontaneous Ca²⁺ oscillation" and spontaneous Ca²⁺ transients with "large spontaneous Ca²⁺ oscillation" (**page 11, lines 30-37**).

In panel B, the results seem to indicate that isoproterenol allows the KO to compensate completely (not "to some extent") for the reduced SR uptake rates.

We agree with the reviewer. The statistics confirm no difference in the mechanical restitution curve of the *Obscn* KO under isoproterenol compared to the WT under the same condition. This indicates therefore that

isoproterenol allows the *Obscn* KO to compensate completely for the reduced SR uptake rates. We changed this part of the text accordingly (**page 11, lines 15-37**).

6. Fig. 6: Wouldn't diastolic Ca²⁺ levels in KO be expected to be elevated even at low pacing rates, if RyR2 leak is involved? Or is the myopathy associated with an increased sensitivity of the RyR2 to Ca²⁺ but only when the levels are increased by stimulation?

We concur with reviewer's assertion that the elevation of diastolic calcium is compatible with an increased RyR2 sensitivity to luminal calcium, which is elevated under all conditions. The result in Figure 6B also suggests that, at low pacing frequency, the threshold of SR calcium content to induce a spontaneous opening of RyR2 channels is low. However, when the pacing frequency is increased, resulting in elevated SR luminal calcium levels, the probability of RyR2 opening increases, as does the diastolic calcium level. This finding suggests that interventions that lead to elevated SR luminal calcium levels, such as high pacing frequency, may stimulate SR calcium leak and thereby increase diastolic calcium levels. This point has been further elaborated in the revised Discussion section (**page 16, lines 7-27**).

It seems doubtful that the method in Panel C is sufficient to conclude that the morphology of the t-tubule system is normal. What do the green and red in Fig. 6C signify? The images seem quite different, with many more and larger discontinuities in the t-system in the KO than in WT. If so, could these help explain some of the results?

We thank the reviewer for this comment which allowed us to identify important results by electron microscopy. The data previously reported in Figure 6C-D aimed at verifying whether there were changes in the number of TT. We agree with the reviewer that a frequent issue when analyzing density is that automatic thresholding in image-processing is sensitive to the overall signal from T-tubules, which can bias threshold settings toward lower values in cells with fewer t-tubules. Using the standardized AutoTT plugin we solved some of these limitations (<https://doi.org/10.1016/j.bpj.2014.05.013>). The parameter originally reported in Figure 6C of the present work is TT-power (A.U.), an index of global density of TATS profiles where transverse and axial components were not discriminated separately. New representative images acquired by confocal microscopy and the TT power are now reported in the revised Figure 2 (panels D and E). The text (**page 9, lines 27-40**) and the legend of Figure 2 (**page 25, lines 14-29**) have been revised, accordingly.

7. The SR in KO skeletal muscle is reduced compared to WT. Why was it not to examined here? A reduced SR compartment would explain many of the results related to Ca²⁺ uptake, ATPase activity, recycling, etc.,

We thank the reviewer for this suggestion. We performed EM analyses on 8-month-old male animals. This resulted in the evidence that *Obscn* KO mice exhibit a significant reduction in longitudinal sarcoplasmic reticulum, accompanied by a noticeable swelling of the T-tubules. These ultrastructural alterations appear to support the biochemical result related to the decrease in SERCA and the altered calcium homeostasis. Representative images from these new electron microscopy experiments and the quantifications of I-SR extension and T-tubule area are now included in the new panels F, G, and H of Figure 2. The methodological details related to these new experiments have been added to the Materials and Methods (**page 7, lines 25-33**). Also the Results and Discussion sections have been revised in light of these new experiments (**page 9, lines 27-40; page 12, lines 2-4; page 13, lines 25-29; page 14, lines 42-45; page 15, lines 1-10 and 28-41; page 16, line 36**).

8. Discussion: The discussion of DCM did not consider the possibility that, had the mice been aged further, they would have developed a classic DCM phenotype.

All the experiments reported were performed with mice at 8 months of age. We agree with the reviewer that the reported phenotype corresponds to a milder form of DCM. Indeed, the decrease in contractility is mild, but is compensated by an increase in end-diastolic volume (EDV), resulting in a compensated DCM phenotype. It is reasonable to hypothesize that if the mice had been subjected to a longer period of aging, they would have developed a classical DCM phenotype. We added this comment to the conclusion (**page 16, lines 40-43**).

The authors mention that there is no difference in Ca²⁺-ATPase, but the data in Table 1 indicate that there is a difference of ~15%. This may not be statistically significant, but given the variability (e.g., Fig. 4), this is not surprising. In fact, as just noted, a difference in enzyme activity could account for many of the results here.

We apologize if this part of Results was not clear enough. The maximal ATPase reported in Table 1 refers to the actomyosin ATPase activity during tension generation at maximal activating calcium. Cardiac muscle strips are demembranated with loss of sarcolemma and membrane organelles (including SR). These experiments are designed to measure the sarcomere energy cost of tension generation (actomyosin ATPase activity/tension; tension cost). While there is some variability in the maximal ATPase activity of the WT and mutant sarcomeres, as outlined by the reviewer, there is no statistically significant difference between the two groups of preparations. Therefore, there is no change in tension cost, which is a remarkable result considering the OBSCN mutation. While the direct molecular effects of DCM-associated genetic variants in Titin and Troponins as well as dystrophin or SR proteins translate into altered myofilament contractile function, here we found that sarcomere function is preserved in *Obscn* KO mice. To avoid misunderstanding we have clarified this in Table 1.

".., likely due to a higher SR Ca²⁺ uptake rate" It seems contradictory to argue that the Ca²⁺ uptake rate is elevated when SERCA levels and ATPase activities are reduced, diastolic Ca²⁺ levels are high, etc.

We agree with the reviewer that the discussion of this aspect in the original version of the manuscript may have caused confusion for the reader. Ultrastructural analysis (new Figure 2F-G) now indicate that the absence of Obscurin leads to SR collapse and volume reduction. This is confirmed by specific Western blot analysis, which showed a decrease in the total SERCA protein levels in *Obscn* KO myocardium (Figure 4B-C). Additionally, the overall amount of PLB protein was reduced, while its phosphorylation at Ser-16 was elevated (Figure 4 B-C). These findings suggest that in *Obscn* KO myocardium under baseline conditions, the diminished inhibition of SERCA by PLB, due to its lower levels and increased phosphorylation, may functionally compensate for the reduced expression of the pump. However, the heightened baseline phosphorylation of PLB at the PKA site (Ser-16), along with the overall reduction in SERCA and PLB proteins, may explain the impaired enhancement of SERCA activity during β -adrenergic stimulation (Iso) (**page 15, lines 28-41**). An additional contributor to the blunted positive limb of the Force-frequency response could be the SR instability and its additional impairment under β -adrenergic stress, due to RyR2 hyperphosphorylation by PKA. We have modified the relevant section of the discussion to better clarify the interpretation of the results compared to the original version (**page 16, lines 2-5**).

In addition, two mechanisms, at least, may explain why *Obscn* KO cardiomyocytes exhibit more spontaneous SR Ca²⁺ release than controls. One can be simply related to the sustained elevation of diastolic Ca²⁺ (Figure 5) in *Obscn* KO cardiomyocytes. Secondly, prolonged exposure to elevated intracellular Ca²⁺ levels could lead, either directly or through activation by CaMKII, to an increased probability of ryanodine receptor (RyR2)

opening and, consequently, to increased spontaneous SR Ca²⁺ release. A third contribution is related to the increased baseline phosphorylation of PLB at the PKA site, which accelerates SERCA activity and thus SR refilling in *Obscn* KO cardiomyocytes. This, in turn, exerts an influence on RyR2 recovery through luminal control. The mechanical restitution data align with this interpretation.

We now added this part to paragraph 'SR instability and arrhythmias' that has been extensively revised (**page 16, lines 7-27**).

The discussion might include a more integrated consideration of the interplay between the changes in Ca²⁺ storage and Ca²⁺ leak that they observe.

We thank the reviewer for all the comments. We believe that the manuscript has now been improved with more integration between the changes in Ca²⁺ storage and Ca²⁺ leakage (**Discussion, pages 14-16**).

Minor:

1. The English could use some editing for typographical errors, word choice (e.g., better terms for "layout", "set up", "dilatative", "heighted", "morpho-functional", etc.) and brevity.

We corrected. Only "Morpho-functional" was not changed as this term is often used in physiology

2. Order of references: The authors might consider putting these in alphabetical order or by year of publication. As is, the order seems random.

The order of references follows the guidelines of the journal.

Reviewer #2 (Comments to the Authors):

This manuscript characterizes the cardiac phenotype of Obscurin knockout mice. The mice were originally published in 2009, but that study focused on skeletal muscle, so it is good to see that finally the heart is studied more closely. The careful analysis shows that while the mice do not have much of a phenotype at baseline, they lack contractile reserve upon stress, e.g. beta adrenergic response is completely absent. The authors then go on to perform a thorough characterization of the ECC phenotype and find Ca leakage and some dysregulation of SERCA and phospholamban expression and phosphorylation.

The work is a careful analysis of the functional phenotype of Obscurin knockout hearts and shows that while there is nothing massively dramatic going on, these mice might be poised to respond with cardiac complications in a stress setting (physical exercise; pregnancy; age). From the Material & Methods it appears that all the studies were done in 2 months old animals; is this really the case? That strikes me as a bit young, since with mouse models phenotypes remain hidden for months; do the authors have any insights how ageing affects the animals? Along a similar line, only once it is mentioned that male hearts were used. Many mouse studies have now shown differences in the response between males and females (which is even more apparent in human patients); can the authors confirm that all the investigations were done in males, or tease apart data for males and females if possible? Also mention in the figure legend (e.g. Figure 1) which sex was used? Both these aspects do not necessarily warrant new experiments in this instance but should be expanded on in the manuscript.

We apologize for the mistake. All experiments were performed using 8-months-old male mice, as stated in the first paragraph of the Methods section. We believe the legitimate doubt raised by reviewer 2 stems from the fact that in the subsequent paragraph ("in vivo studies"), we mistakenly reported that the

experiments were performed on 8-weeks-old animals instead of 8-months-old, and that the sex of the animals used was not specified in the other parts of the manuscript.

The error in the "in vivo studies" paragraph has been corrected, and the sex and age of the animals used in the experiments are now reported also in the figure legends.

I am also wondering whether there is additional electron microscopy data available on the heart. The Lange et al., 2009 paper only shows zoom ins on individual skeletal muscle sarcomeres and since the authors claim here that they see a compensated DCM phenotype, it would be really interesting to see whether there are less distinct M-bands and more convoluted intercalated discs in these hearts, cellular phenotypes of full blown DCM seen by others (e.g. Schoenauer et al., 2011; PMID 21069531; Ehler et al., 2001; PMID 11352937).

We agree with the reviewer. Indeed, the relevance of the ultrastructural analysis of cardiomyocytes in KO hearts has also been emphasized by reviewer 1. We have therefore performed electron microscopy analyses on 8-month-old male animals. Importantly, we found a significant reduction in longitudinal sarcoplasmic reticulum extension, accompanied by noticeable swelling of the T-tubules in Obscn KO cardiomyocytes. These ultrastructural alterations appear to support both the biochemical result related to the decrease in SERCA protein levels and the functional data highlighting disturbances in calcium homeostasis. Representative images from these electron microscopy experiments and the quantifications of I-SR extension and T-tubule area are now included in the new panels F, G, and H of Figure 2. The methodological details related to these new experiments have been included in the Materials and Methods sections (**page 7, lines 25-33**). Also the Results and Discussion sections have been revised in light of these new experiments (**page 9, lines 27-40; page 12, lines 2-4; page 13, lines 25-29; page 14, lines 42-45; page 15, lines 1-10 and 28-41; page 16, line 36**).

Otherwise there are just minor errors that need addressing:

- 1. There is a glitch with the citations (three authors cited in the text and the like), which needs to be fixed.**
- 2. Abstract: Obscurin is -a- giant sarcomeric**
- 3. page 3: signaling domains in obscurin: are at the C-terminus not at the N-terminus as stated here**
- 4. page 5: Noureddine and Gehmlich - then there is a gap - something missing here?**
- 5. page 6: dilated not dilatative**

We thank the reviewer for noticing all these minor errors that have been now corrected in the revised version of the manuscript.

6. Figure 2C: shows what? Information missing from the figure legend; first time it is mentioned is in the Discussion...

We apologize for the oversight in the assembly of the manuscript. The immunofluorescence experiment to assess the presence of collagen III is now described in the "Results" section (**page 9, lines 28-30**) and included in the legend of Figure 2 (**page 25, lines 15-29**).

7. The differences in SERCA and Plb expression look very noisy in the immunoblots; much noisier than the data in the quantification; again I am wondering whether a male/female difference might be at the root of this?

We agree with Reviewer 2. This concern was also raised by both Reviewer 1 and 3. Accordingly, we performed new western blot experiments to assess the expression levels of SERCA, PLB, and p-PLB, preparing new lysates from 5 WT hearts and 5 KO hearts taken from 8-month-old animals.

The results confirmed that the levels of SERCA and even more of PLB are significantly reduced in Obscn KO mice compared to WT animals, while the inhibitory phosphorylation on PLB at serine 16 is significantly increased in KO animals compared to WT.

Revised Panels B and C of Figure 4 and the corresponding legend (**page 25, lines 42-48 and page 27, lines 1-2**) now report these new experiments. The "Western blot analysis" section in the Materials and Methods has also been accordingly modified (**page 7, lines 35-45, and page 8, lines 1-13**).

8. Figure 6: Actually the T-tubules do look a bit more purpley in the Obscurin KO cardiomyocytes, so there might be a phenotype that is not picked up by the analysis they performed; ImageJ does not solve every problem. Try the algorithm that Candamamy et al., 2014; PMID 24280220; used for their analysis?

As also stated in our reply to reviewer 1, the data previously reported in Figure 6C-D aimed at verifying whether there were changes in the number, density and organization of TTs. We agree with the reviewer that the images originally presented were not fully explanatory of the experimental outcome. However, by using a standardized AutoTT plugin (<https://doi.org/10.1016/j.bpj.2014.05.013>), we improved the outcome of this analysis. Accordingly, we replaced the images reported in submitted Figure 6C with images that are more representative of the statistics obtained from all whole analysis. In addition, we substituted "T-element density" with "TT power A.U." New representative images acquired by confocal microscopy and the TT power, are now presented in the revised Figure 2 (panels D and E), where we believe that this experiment, in light of the new data obtained from the electron microscopy analysis, find a more logical placement within the manuscript. The text (**page 9, lines 27-40**) and the legend of Figure 2 (**page 25, lines 15-29**) have been revised, accordingly.

9. page 21: In this context it is noteworthy that - gap- observed - who?

In the original text, a "was" was missing. We apologize for this. However, this part has been revised in light of the results obtained from the electron microscopy experiments (**page 14, lines 27-45 and page 15, lines 1-10**)

10. page 22: line 2 from bottom: Obscn KO not OBSC KO (assuming we are still talking about mice...)

This has been corrected.

11. The discussion could do with some trimming; it is a bit long at the moment.

The discussion has been revised to reflect the changes made to the text, and shortened as suggested by the reviewer.

Reviewer #3 (Comments to the Authors):

The authors use a obscurin KO mouse model to investigate the role of this protein in cardiac function. They show that dilation of the heart takes place with reduced contractility. This is not due to myofilament dysfunction but rather changes in calcium handling. Also, a notable blunting of beta1-adrenergic receptor stimulation is seen.

I think that the paper is of interest and that most of the experiments are performed to a high standard. I do have a number of questions and comments that I hope would improve the paper.

1) The lack of increase in force after isoprenaline in the KO mice is intriguing. I think this is one of the key

findings of the paper. Unfortunately the mechanism of this is not explored as elaborately as I would have liked it to be. All follow-up experiments do not look at the effects on PLB phosphorylation or Ca-transient kinetics do not study the effect after ISO. I think this is key to understanding the mechanism.

Is beta1-AR signaling still intact? Can it phosphorylate cTnI or cMyBP-C but not PLB?

This is indeed a central part of the mechanism we have described. To the best of our knowledge, in the heart the β 1-receptor contributes to vasorelaxation and the β 2-receptor to cardiac contractility. Although the predominant cardiac β 1 subtype (\approx 70% to 80% depending on species) is the most potent stimulator of cardiac function, expression levels are quite small: no more than 50 to 70 fmol/mg membrane protein in most species. Thus, there is little receptor reserve.

As we discussed: “Under Iso (compared to baseline) the acceleration of relaxation was impaired in Obscn KO. In particular, at 8 Hz, the RT50% under Iso did not show a significant reduction compared to baseline. This means that velocities do change in both models, but to a lesser degree in the Obscn KO. This evidence supports an impairment of the β -adrenergic response, which is consistent with the loss of SR integrity. Therefore, in Obscn-KO ultrastructural SR collapse and volume reduction was confirmed by western blot analysis, which showed a decrease in the total SERCA protein levels in Obscn KO myocardium (Figure 3B-C). Additionally, the overall amount of an (PLB) protein was reduced, while its phosphorylation at Ser-16 was elevated (Figure 3C). These findings suggest that in Obscn KO myocardium under baseline conditions, the diminished inhibition of SERCA by PLB, due to its lower levels and increased phosphorylation, may compensate functionally for the reduced SR volume and, therefore, expression of the pump. However, the increased baseline phosphorylation of PLB at the PKA site (Ser-16), along with the overall reduction in SERCA and PLB proteins, could be an additional explanation to the impaired enhancement of SERCA activity during β -adrenergic stimulation (i.e. Iso). These data suggest that the observed multifunction of the β -adrenergic response in Obscn KO mice relay on the loss of SR volume and, as a consequence, on the dysregulation of the intracellular pathway rather than (or in addition to) the receptor expression levels itself or other targets of PKA-mediated phosphorylations (cTnI or cMyBP-C) (**page 15, lines 38-41**). This last point differs from the data reported by Fujita et al. (Fujita K. et al., 2022 Preprint), because in that study the Obscn KO model shows a perfectly adequate response to beta-adrenergic stimulation (with dopamine) in vivo. However, it should be noted that our experiments were performed on denervated hearts, and thus the lack of an adequate positive inotropic response observed ex vivo could be compensated in vivo by a different adrenergic tone or an increased density of sympathetic terminals. A limitation of the current study is the lack of measurement of beta receptor density in the myocardium of our mouse models. An additional contributor to the blunted positive limb of the force-frequency response could be the instability of the residual SR volume and its additional impairment under β - adrenergic stress, due to RyR2 hyper-phosphorylation by PKA. This latter sentence was added to the discussion (**page 16, lines 2-5**).

2) The Western blot data of SERCA and PLB in figure 4 is not of sufficient quality to ensure reliable quantification. PLB seems missing in most samples (even in WT), SERCA expression is highly variable. This has to be improved, because the interpretation of the results depend on it. The results showing higher PLB-Ser16 phosphorylation is entirely driven by a low PLB level, not a high Ser16 phosphorylation.

This concern was also raised by both Reviewer 1 and 2. Hence we performed new WB experiments using freshly prepared heart lysates from 5 WT hearts and 5 KO hearts taken from 8-month-old animals. The results obtained indicate that the levels of SERCA and PLB are significantly reduced in Obscn KO mice compared to WT animals, and that the levels of serine 16 phosphorylation of PLB are significantly increased in KO animals compared to WT, in agreement with what we previously reported.

Revised Panels B and C of Figure 4 and the corresponding legend (**page 25, lines 42-48 and page 27, lines 1-2**) now report these new experiments. The "Western blot analysis" section in the Materials and Methods has also been accordingly modified (**page 7, lines 35-45, and page 8, lines 1-13**).

3) In Figure 3 you show that forces do not change, but that time to peak and time to baseline do change with ISO in the KO mice. Does that mean that velocities do change in both models, but to a lesser degree in the KO?

We thank the reviewer for this comment. The time course of twitch contractions at baseline was preserved in Obscn-KO. Under Iso (compared to baseline) the acceleration of relaxation was impaired in Obscn-KO. In particular, at 8 Hz, the RT50% under Iso did not show a significant reduction compared to baseline. This means that velocities do change in both models, but to a lesser degree in the KO. This latter point has been added in the discussion (**page 15, lines 26-31**).

4) Figure 2C is not described in the paper nor the legend. There is also no quantification provided

This was an oversight on our part, for which we apologize. The analysis of type III collagen expression is now described in the "Results" sections (**page 9, lines 28-30**), and in the legend of Panel C of Figure 2 (**page 25, lines 15-29**). Since the levels of Type III collagen are just about levels of detection, no quantification has been performed.

5) The intro is well written but too long

The introduction has been significantly shortened.

Minor:

1) The symbols in figure 3B do not match the legend for the KO group.

We apologize for the oversight. Fig. 3 has been corrected.

2) In the text it is stated that the mice have a heart rate of 8-9 Hz. This is typically presented as bpm in in vivo experiments.

We changed as follows: "The mean spontaneous heart rate in sinus rhythm during echocardiography was 480-540 bpm (8-9 Hz) with no differences between the two groups" (**page 9, lines 5-7**).

3) On page 7 you describe your animals as transgenic. These are not transgenic, they are knock-out animals. Also in that section you say heighted instead of weighed.

We thank the reviewer for this observation, and we apologize for the mistake. We have now corrected the two sentences (**page 5, lines 38-40**).

April 3, 2025

Prof. Vincenzo Sorrentino
University of Siena
Molecular and Developmental Medicine
via Aldo Moro 2
Siena 53100
Italy

Re: 202413696R1

Dear Prof. Sorrentino,

I am pleased to let you know that your manuscript, entitled "Obscurin deficiency leads to compensated dilated cardiomyopathy with reduced inotropic reserve and increased arrhythmias" is scientifically acceptable for publication in Journal of General Physiology. Formal acceptance will follow when it is modified in accordance with our editorial policies.

Please note items that need attention are listed at the bottom of this email (under 'manuscript formatting checklist') and on the attached marked-up pdf file. Please also be sure to include a letter addressing the reviewers' comments point-by-point (if applicable) and a copy of the text with alterations highlighted (boldfaced or underlined). Your manuscript should be a double-spaced MS Word file and include editable tables, if appropriate.

Lastly, JGP requires a data availability statement for all research article submissions. These statements will be published in the article directly above the Acknowledgments. The statement should address all data underlying the research presented in the manuscript. Please visit the JGP instructions for authors for guidelines and examples of statements at <https://rupress.org/jgp/pages/editorial-policies#data-availability-statement>.

Please submit your final files via this link:
Link Not Available

Thank you for choosing to publish your research in JGP and please feel free to contact me with any questions.

Sincerely,

Henk L. Granzier, Ph.D.
On behalf of Journal of General Physiology

Journal of General Physiology's mission is to publish mechanistic and quantitative molecular and cellular physiology of the highest quality; to provide a best in class author experience; and to nurture future generations of independent researchers.

Manuscript formatting checklist:

- As mentioned in your previous decision letter, JHI requires authors to submit Source Data used to generate figures containing gels and Western blots with all revised manuscripts. This Source Data consists of fully uncropped and unprocessed images for each gel/blot displayed in the main and supplemental figures. Since your paper includes cropped gel and/or blot images, please be sure to provide one Source Data file for each figure that contains gels and/or blots along with your revised manuscript files. File names for Source Data figures should be alphanumeric without any spaces or special characters (i.e., SourceDataF#, where F# refers to the associated main figure number or SourceDataFS# for those associated with Supplementary figures). The lanes of the gels/blots should be labeled as they are in the associated figure, the place where cropping was applied should be marked (with a box), and molecular weight/size standards should be labeled wherever possible.
- MS Word document of text needed (including editable tables)
- MS Word document of supplemental text needed, if applicable (including figure legends and editable tables)
- Brief Statement describing supplementary information needed, if applicable (in subsection at end of Materials & Methods)
- Please include a data availability statement preceding the Acknowledgments section. Please see <https://rupress.org/jgp/pages/editorial-policies#data-availability-statement>
- Figures created at sufficient resolution and in acceptable format (including supplemental if applicable). If working in Illustrator, we prefer .ai or .eps file format. If working in Photoshop please use 600dpi/1000dpi .tiff or .psd file format. Minimum resolution at estimated print size: Minimum resolution for all figures is 600 dpi. For figures that contain both photographs and line art or text, 600 dpi is highly recommended. Figures containing only black and white elements (line art, no color, and no gray) should be 1,000 dpi. Maximum figure size is 7 in wide x 9 in high (17.5 x 22.8 cm) at the correct resolution. <https://jgp.rupress.org/fig-vid->

guidelines

- Supplemental figures, if any, conforming to same guidelines as manuscript figures (noted above)
 - If images resemble one from a prior publications, the author must seek permissions (to reproduce or adapt) from the original publisher. [You can resubmit your paper while waiting to hear back from the original publisher but please keep us updated]
 - All authors must complete a disclosure form prior to acceptance. A link to complete the form has been sent to all coauthors. Please provide the editorial office with updated email addresses if necessary
-

Reviewer #1 (Comments to the Authors):

The authors have addressed all my previous comments with great care. I have no further concerns.

Reviewer #2 (Comments to the Authors):

Most of my questions were answered by the revised version.

Reviewer #3 (Comments to the Authors):

I am happy with the answers the authors provided.